# High performance platinum single atom electrocatalyst for oxygen reduction reaction

Jing Liu[1,2,*], Menggai Jiao[2,3,*], Lanlu Lu[4], Heather M. Barkholtz[5], Yuping Li[6], Ying Wang[7], Luhua Jiang[6], Zhijian Wu[3], Di-jia Liu[5], Lin Zhuang[7], Chao Ma[8], Jie Zeng[8], Bingsen Zhang[9], Dangsheng Su[9,10], Ping Song[1], Wei Xing[1], Weilin Xu[1], Ying Wang[2], Zheng Jiang[4] & Gongquan Sun[6]

For the large-scale sustainable implementation of polymer electrolyte membrane fuel cells in vehicles, high-performance electrocatalysts with low platinum consumption are desirable for use as cathode material during the oxygen reduction reaction in fuel cells. Here we report a carbon black-supported cost-effective, efficient and durable platinum single-atom electro-catalyst with carbon monoxide/methanol tolerance for the cathodic oxygen reduction reaction. The acidic single-cell with such a catalyst as cathode delivers high performance, with power density up to $680\,mW\,cm^{-2}$ at 80 °C with a low platinum loading of $0.09\,mg_{Pt}\,cm^{-2}$, corresponding to a platinum utilization of $0.13\,g_{Pt}\,kW^{-1}$ in the fuel cell. Good fuel cell durability is also observed. Theoretical calculations reveal that the main effective sites on such platinum single-atom electrocatalysts are single-pyridinic-nitrogen-atom-anchored single-platinum-atom centres, which are tolerant to carbon monoxide/methanol, but highly active for the oxygen reduction reaction.

[1] State Key Laboratory of Electroanalytical Chemistry, Jilin Province Key Laboratory of Low Carbon Chemical Power, Changchun Institute of Applied Chemistry, Chinese Academy of Sciences, 5625 Renmin Street, Changchun 130022, China. [2] University of Chinese Academy of Sciences, Beijing 100049, China. [3] State Key Laboratory of Rare Earth Resource Utilization, Changchun Institute of Applied Chemistry, Chinese Academy of Sciences, Changchun 130022, China. [4] Shanghai Synchrotron Radiation Facility, Shanghai Institute of Applied Physics, Chinese Academy of Sciences, Shanghai 201204, China. [5] Chemical Sciences and Engineering Division, Argonne National Laboratory, Argonne, Illinois 60439, USA. [6] Division of Fuel Cell and Battery, Dalian National Laboratory for Clean Energy, Dalian Institute of Chemical Physics, Chinese Academy of Sciences, Dalian 116023, China. [7] College of Chemistry and Molecular Sciences, Hubei Key Lab of Electrochemical Power Sources, Wuhan University, Wuhan 430072, China. [8] Department of Chemical Physics, Hefei National Laboratory for Physical Sciences at the Microscale, University of Science and Technology of China, Hefei, Anhui 230026, China. [9] Shenyang National Laboratory for Materials Science, Institute of Metal Research, Chinese Academy of Sciences, Shenyang 110016, China. [10] Fritz Haber Institute of the Max Planck Society, Faradayweg 4–6, Berlin 14195, Germany. * These authors contributed equally to this work. Correspondence and requests for materials should be addressed to W.X. (email: weilinxu@ciac.ac.cn) or to Y.W. (email: ywang_2012@ciac.ac.cn) or to Z.J. (email: jiangzheng@sinap.ac.cn) or to G.S. (email: gqsun@dicp.ac.cn).

The electrochemical oxygen reduction reaction (ORR) is the limiting half-reaction for low-temperature fuel cells, and currently costly carbon-supported platinum (Pt) nanoparticle (NP)-based electrocatalysts (Pt/C, with Pt loading up to 60 wt %) are used extensively to generate adequate rates[1–3]. At the moment, typical Pt-NP-based electrocatalysts represent about half the cost (projected by the US Department of Energy) of an automotive fuel cell stack[4], which hampers the commercialization of fuel cell technology. Intrinsically, the high cost of conventional Pt-NP-based catalysts or the large consumption of Pt in fuel cells is mainly due to the sluggish ORR kinetics and the low Pt utilization efficiency on a per Pt atom basis since only a small portion of Pt atoms on the particle surface are involved in catalysis[5]. In principle, downsizing the Pt NPs to single atoms presents one of the most effective ways to reduce the cost of Pt catalysts in fuel cells by enhancing the Pt utilization efficiency. It has been realized that the single-atom catalysts (SAC) are more reactive than metal clusters or particles in some cases[5–10]. As for the Pt application in fuel cells, ideally, a maximum Pt utilization or solution to the cost problem of conventional Pt/C is to prepare high-efficient carbon-supported Pt SAC, in which all the individual Pt atoms are involved in catalysis with utilization efficiency in 100% (refs 10–14). To our knowledge, due to the fact that only the traditional Pt-NP-based Pt/C with high Pt content can show practical activity for ORR in fuel cells, there is no report describing the efficient electrochemical four-electron (4e) ORR on carbon-supported Pt SAC for fuel cells, although Pt single atoms supported on different supports have been studied for some other reactions, such as CO/NO oxidation, the hydrogenation of nitroarenes[10–14] and the production of hydrogen peroxide from oxygen[15]. Ideally, if carbon-supported Pt SACs can be utilized successfully in fuel cells for 4e ORR, the cost of high-efficient Pt SACs will not be a hindrance anymore for the commercialization of fuel cells due to the extremely low Pt consumption.

In the present work, we report a cost-effective, high-performance and durable carbon-supported Pt SAC for highly efficient 4e ORR in fuel cells with high Pt utilization of $0.13 \, g_{Pt} \, kW^{-1}$. Density functional theory (DFT) calculations indicate that the single-pyridinic-nitrogen(P-N)-atom-anchored single Pt atom centres are the main active sites, which are highly active for ORR, but are tolerant to CO/methanol.

## Results

**Physical characterization of carbon-supported Pt SAC**. The optimal carbon-supported doped-N triggered Pt SAC (denoted as $Pt_1$-N/BP, with a Pt loading of 0.4 wt % from inductively coupled plasma mass spectrometry (ICP-MS) and N 2.7 wt % from elementary analysis) was obtained based on a simple optimization procedure with cheap carbon black (BP 2,000 (BP) with large surface area $(1,391.3 \, m^2 \, g^{-1}))$ as support, urea and chloroplatinic acid $(H_2PtCl_6 \cdot H_2O)$ as N and Pt precursors, respectively (see Methods section, Supplementary Figs 1–3). A pure N-doped carbon (denoted as N/BP) and a pure carbon-supported Pt SAC with Pt 0.4 wt % (denoted as $Pt_1$/BP) were also prepared and characterized for comparison. Individual heavy Pt atoms on carbon support can be discerned in the atomic-resolution high-angle annular dark-field (HAADF) images through sub-Ångström resolution, aberration-corrected scanning transmission electron microscopy (STEM)[11,16,17]. For $Pt_1$-N/BP, Fig. 1a clearly shows individual Pt atoms (bright spots) uniformly dispersed on the surface of carbon. Examination of multiple regions reveals that only individual Pt atoms are present in $Pt_1$-N/BP. Additional low-magnified HAADF-STEM images do not show any Pt NPs or clusters in this catalyst (Supplementary

Fig. 4). When there is no N-doping on carbon, interestingly, as shown in Fig. 1b and Supplementary Fig. 4, besides the individual Pt atoms, Pt NPs are also observed occasionally on $Pt_1$/BP with the same Pt loading of 0.4 wt % as that on $Pt_1$-N/BP. The difference indicates that the doped N on carbon can anchor the Pt single atoms and prevent their aggregation[18].

Furthermore, the N 1s X-ray photoelectron spectroscopic (XPS) spectrum of $Pt_1$-N/BP (Fig. 1c) shows five different bonding configurations of N atoms. Besides the well-known four types of N, pyridine-like (P-N, 398.3 eV, 42.9%), pyrrole-like (Py-N, 399.9 eV, 24.3%), graphitic (G-N, 400.1 eV, 15.7%) and oxidized (O-N, 403.3 eV, 7.3%) nitrogen, interestingly, similar to the formation of Fe-N bonding observed on $Fe-N_x/C$ catalysts[19,20], Pt-N bonding (399.2 eV, 8.1%) was also observed[21], indicating a strong interaction between single Pt atoms and the neighbouring doped-N due to the anchoring effect of doped-N to individual Pt atoms observed from above HAADF images. As for the Pt 4f XPS data for $Pt_1$-N/BP, as shown in Fig. 1d, most of the Pt atoms (86.6%) on it are metal Pt(0), only a small amount of them were oxidized as PtO. However, as for the $Pt_1$/BP without N-doping, Fig. 1e shows that most of the Pt atoms (69.8%) are oxidized as PtO. The big difference observed here indicates that the strong interaction between Pt atoms and doped-N can hugely prohibit the oxidation of Pt atoms by oxygen in air. Compared with $Pt_1$/BP, the Pt 4f peaks on $Pt_1$-N/BP show positive shifts due to the well-known strong interaction between doped-N and Pt atoms[22].

As shown in Supplementary Fig. 5, X-ray diffraction patterns of sample $Pt_1$-N/BP did not show any Pt-containing crystal phases, primarily because of the insensitivity of X-ray diffraction to single Pt atoms (Fig. 1a). To further verify that $Pt_1$-N/BP contains only atomically dispersed individual Pt atoms, X-ray absorption of fine structure (XAFS) spectra were obtained for both $Pt_1$-N/BP and $Pt_1$/BP.

XAFS can probe the local atomic and electronic structure of absorbing atoms. XAFS is typically divided into X-ray absorption near edge structure, which provides information primarily about geometry and oxidation state, and extended XAFS (EXAFS) which provides information about metal-site ligation[23]. Pt $L_3$-edge XAFS of the samples ($Pt_1$-N/BP and $Pt_1$/BP) and reference (Pt foil and bulk $PtO_2$) are shown in Fig. 1f,g and Supplementary Fig. 6. The white-line intensities in the normalized X-ray absorption near edge structure spectra reflect the oxidation state of Pt (ref. 24). As shown in Fig. 1f, the white-line intensity of $Pt_1$/BP (without N-doping) is between those of Pt foil and $PtO_2$, indicating the oxidation of some Pt atoms in $Pt_1$/BP. While the white-line intensity of $Pt_1$-N/BP is lower than that of $Pt_1$/BP, indicating that the content of Pt oxide in $Pt_1$-N/BP is lower than that in $Pt_1$/BP[21], consistent with the above XPS results (Fig. 1d,e). These data confirm that the doped-N can prohibit the oxidation of single Pt atoms.

After removal of the background, the total EXAFS functions with $k^2$-weight were extracted from the X-ray absorption spectra for all the specimens as shown in Supplementary Fig. 6. The EXAFS spectra of $Pt_1$-N/BP at the Pt L3-edge are characterized by the absence of oscillations at a high $k$ region of $k > 10$ Å. It indicates the dominance of low Z back scatters, which should be carbon support, nitrogen or oxygen adsorbate in our system. Correspondingly, in the Fourier transforms (R-space, Fig. 1g) of the EXAFS data for $Pt_1$-N/BP, there is only one prominent peak centred around 1.5 Å originated from the Pt-C/N/O contribution, which is an indication that $Pt_1$-N/BP contains only single Pt atoms. In contrast to $Pt_1$-N/BP, the EXAFS spectrum of $Pt_1$/BP with the same Pt content but with no N-doping displays an oscillation at the high $k$ region of $k > 10$ Å and the corresponding R-space has a weak peak at 2.5 Å from the Pt-Pt contributions.

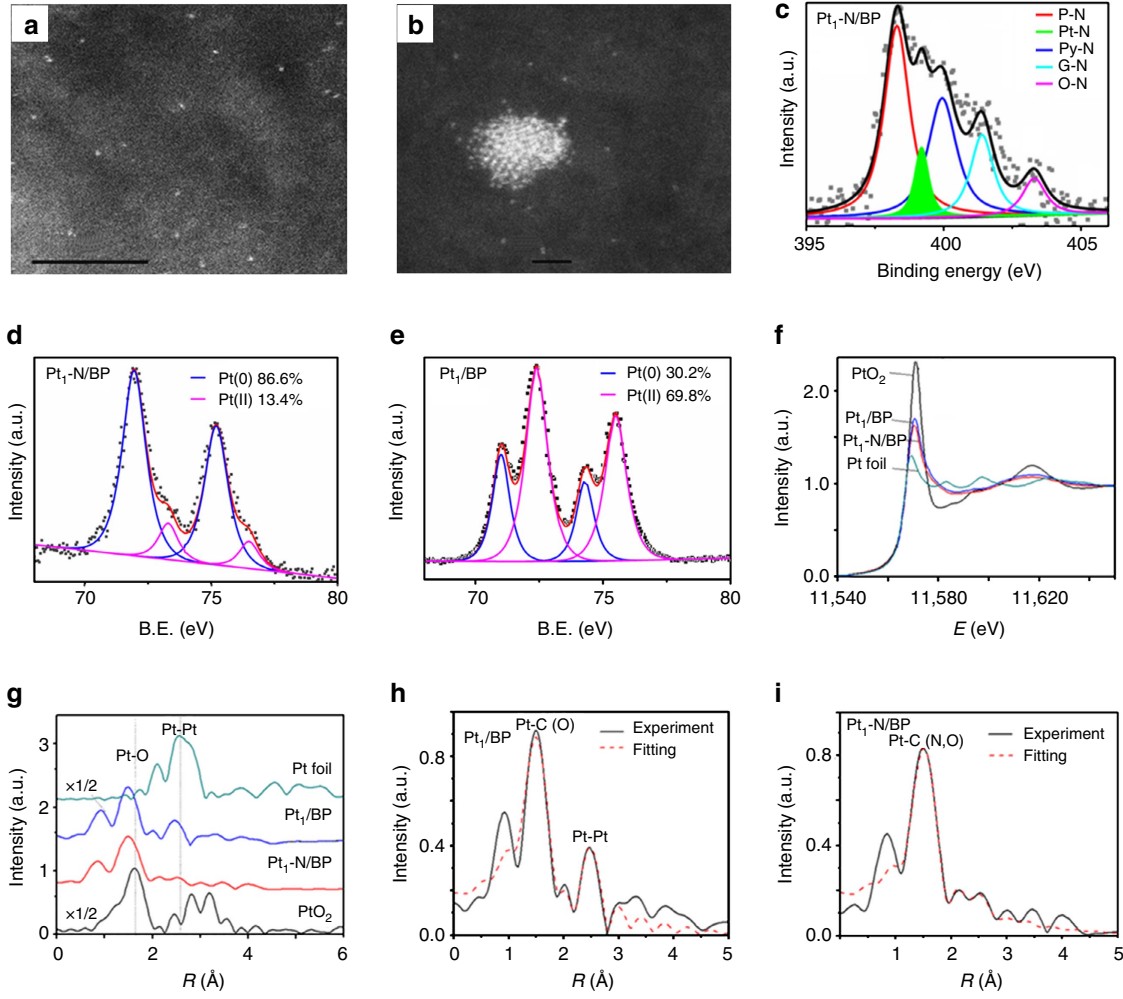

**Figure 1 | Physical characterization of different catalysts.** HAADF-STEM images of $Pt_1$-N/BP (**a**) and $Pt_1$/BP (**b**; corresponding scale bar, 5 nm (**a**) and 1 nm (**b**)). In sample $Pt_1$-N/BP, Pt single atoms (bright spots) are seen to be uniformly dispersed on carbon, while in sample $Pt_1$/BP, besides Pt single atoms, small Pt clusters can be found. (**c**) XPS spectra for N 1s in $Pt_1$-N/BP. (**d,e**) XPS spectra for Pt 4f in $Pt_1$-N/BP (**d**) and $Pt_1$/BP (**e**). (**f**) Pt L3-edge XANES for all the samples. (**g**) The $k^2$-weighted R-space FT spectra from EXAFS. $\Delta k = 3.1$–10.6 Å$^{-1}$ for $Pt_1$-N/BP and $Pt_1$/BP, but $\Delta k = 3.1$–13.8 Å$^{-1}$ for Pt foil and $PtO_2$. EXAFS fitting in R-space for (**h**) $Pt_1$/BP, (**i**) $Pt_1$-N/BP.

**Table 1 | EXAFS parameters of samples $Pt_1$-N/BP and $Pt_1$/BP.**

| Sample | Shell | N | R (Å) | $\sigma^2 \times 10^3$ (Å$^2$) |
|---|---|---|---|---|
| Pt foil | Pt-Pt | 12.0 | 2.77 | |
| $PtO_2$ | Pt-O | 6.0 | 2.07 | |
| | Pt-Pt | 6.0 | 3.10 | |
| $Pt_1$/BP | Pt-C(O) | 5 | 1.98 ± 0.01 | 5.7 ± 1.0 |
| | Pt-Pt | 1.1 | 2.54 ± 0.02 | 2.2 ± 2.0 |
| | Pt-C | 4 | 2.87 ± 0.03 | 10.4 ± 5.5 |
| $Pt_1$-N/BP | Pt-C(O) | 3 | 1.97 (± 0.01) | 2.6 ± 1.0 |
| | Pt-N(O) | 2 | 2.04 (± 0.01) | 5.6 ± 2.1 |
| | Pt-C | 4 | 2.89 (± 0.01) | 11.6 ± 2.4 |

$\sigma^2$, Debye–Waller factor; EXAFS, extended X-ray absorption fine structure; N, coordination number with an error of 20%; R, distance between absorber and backscatter atoms.
Pt foil parameter from data_76153-ICSD; $PtO_2$ parameter from data_24922-ICSD.

It suggests that sample $Pt_1$/BP contains not only single Pt atoms but also small $Pt_x$ clusters or NPs, consistent with the HAADF results shown in Fig. 1a,b.

To extract quantitative structural parameters for the atoms surrounding the central Pt atoms, we have fitted the Fourier transform main peaks from 1.0 to 3.0 Å in the R-space for both samples as shown in Fig. 1h,i. The best fitting values of structural parameters are listed in Table 1. There are low Z backscatter (C, O or N)[25] contributions around a distance of 2.0 Å with a total coordination number of 5 for both samples. For $Pt_1$/BP,

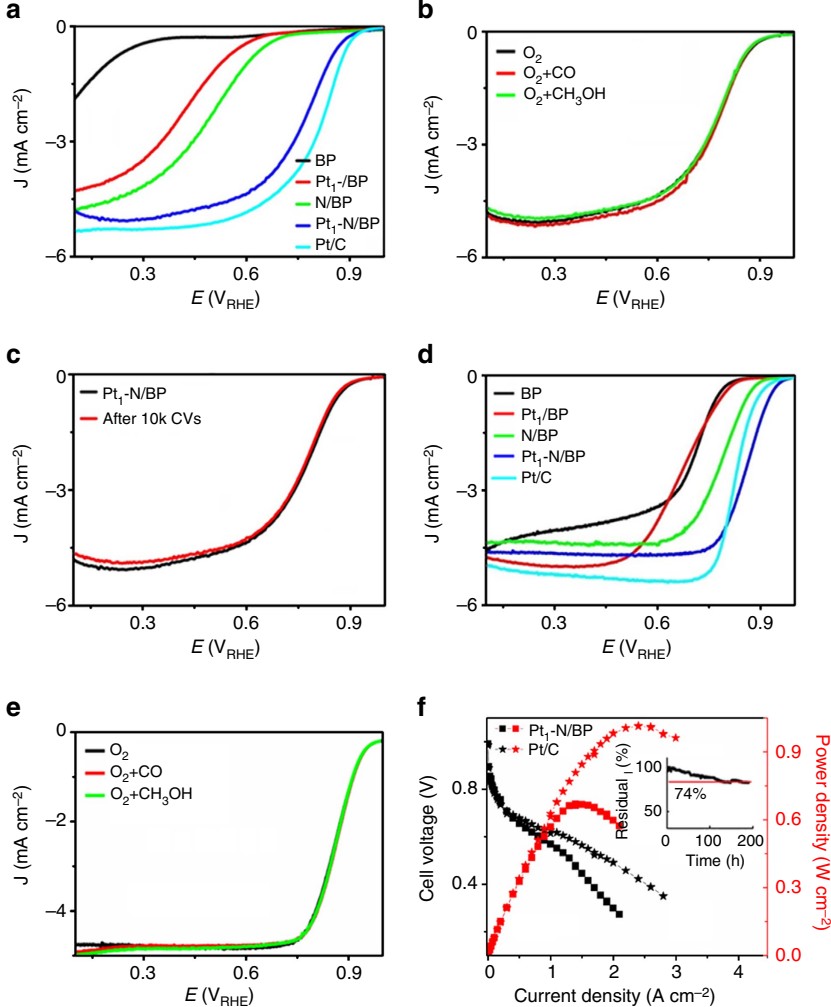

**Figure 2 | Electrochemical characterization of different catalysts.** (**a**) RRDE polarization curves of BP, N/BP, Pt$_1$/BP, Pt$_1$-N/BP and commercial Pt/C in O$_2$-saturated 0.1 M HClO$_4$ with a scan rate of 5 mV s$^{-1}$ and rotation speed of 1,600 r.p.m. (**b**) The tolerance of Pt$_1$-N/BP to CO (saturated) and methanol (0.5 M) in O$_2$-saturated 0.1 M HClO$_4$. (**c**) Long-term operation stability of Pt$_1$-N/BP in O$_2$-saturated 0.1 M HClO$_4$. (**d**) RRDE polarization curves of BP, N/BP, Pt$_1$-BP, Pt$_1$-N/BP and commercial Pt/C in O$_2$-saturated 0.1 M KOH with a scan rate of 5 mV s$^{-1}$ and rotation speed of 1,600 r.p.m. The catalyst loading of Pt-free catalysts is 0.39 mg cm$^{-2}$, the Pt loading of Pt$_1$-BP and Pt$_1$-N/BP is 1.56 μg Pt cm$^{-2}$; the Pt loading of commercial Pt/C is 24 μg Pt cm$^{-2}$. (**e**) The tolerance of Pt$_1$-N/BP to CO (saturated) and methanol (0.5 M) in O$_2$-saturated 0.1 M KOH. (**f**) The voltages and power densities of H$_2$/O$_2$ fuel cells with Pt$_1$-N/BP (cathode: 2.5 mg$_{Pt1-N/BP}$ cm$^{-2}$ or 0.01 mg$_{Pt}$ cm$^{-2}$ (marked with solid squares)) and commercial Pt/C (cathode: 0.2 mg$_{Pt}$ cm$^{-2}$ (marked with stars)) as cathodes in acid (membrane: Nafion212, anode: 0.1 mg$_{Pt}$ cm$^{-2}$ (marked with stars) and 0.08 mg$_{Pt}$ cm$^{-2}$(marked with solid squares), back pressure: 0.2 bar, 80 °C) fuel cells with H$_2$ and O$_2$ in 100% RH. Insert: the acidic fuel cell (with Pt$_1$-N/BP as cathode) lifetime test at 0.5 V and 80 °C for 200 h.

there is a Pt-Pt shell at 2.54 Å with coordination number of 1.1, both of which are much lower than those in bulk platinum (2.78 Å with coordination number of 12)[26]. The significantly shorter Pt-Pt bond distance in small Pt clusters or NPs has been predicted by theoretical studies[27]. From these data, we conclude that Pt$_1$/BP contains single Pt atoms as a major component and also a few small Pt$_x$ clusters or NPs. In contrast to Pt$_1$/BP, the EXAFS fitting in R-space (Table 1) for Pt$_1$-N/BP does not contain any Pt-Pt coordination.

**Rotating ring-disk electrode tests**. To assess the catalytic activity of these catalysts for ORR, we performed linear sweep voltammetry test on a rotating ring-disk electrode (RRDE) in acidic condition (0.1 M HClO$_4$). As Fig. 2a shows, the pure carbon black BP is almost inert for ORR (black curve). As expected, the solo doping of N on BP carbon (N/BP, green curve) can enhance the

ORR activity[28,29], indicated by a high half-wave potential ($E_{1/2}$) of 0.51 V versus reversible hydrogen electrode (RHE; V$_{RHE}$). Compared with N/BP, the pure carbon-supported Pt SAC (Pt$_1$/BP, red curve) shows lower ORR performance with $E_{1/2}$ of 0.44 V$_{RHE}$, indicating the carbon-supported single Pt atoms in oxidation state (Fig. 1e,f) are almost inert to ORR process[15]. That's probably the reason why there is no report before about the application of carbon-supported Pt SACs as ORR electrocatalysts in fuel cells. While significantly, with the codoping of N, the modified Pt SAC (Pt$_1$-N/BP, blue curve) shows much higher catalytic activity for ORR compared with N/BP or Pt$_1$/BP, indicating that the intrinsic catalytic activity of the complex active sites based on doped-N and single Pt atoms is much higher than that of the pure doped-N based active sites or carbon-supported Pt$_1$ sites. This fact further implies that the catalytic activity of Pt single atoms for ORR could be triggered tremendously by the doped-N atoms to a high level due to a

synergetic effect between doped-N and Pt single atoms, gleaned from its much higher $E_{1/2}$ (0.76 $V_{RHE}$, blue curve in Fig. 2a) and a 4e ORR process with a much lower $H_2O_2$ yield compared with N/BP or $Pt_1$/BP (Supplementary Fig. 7). Significantly, the ORR activity of $Pt_1$-N/BP in acid is on the same level as that of the most active Pt-free ORR electrocatalysts in acid (Supplementary Table 1)[30,31] and close to the traditional state-of-art Pt-NP-based Pt/C with much higher Pt loading (cyan curve in Fig. 2a)[32,33].

Interestingly, compared with the high activity of conventional commercial Pt/C for the electro-oxidation of methanol or CO (Supplementary Fig. 8), the $Pt_1$-N/BP shows high tolerance to methanol or CO in $O_2$-saturated solution (Fig. 2b). This result indicates that the doped-N triggered Pt SAC ($Pt_1$-N/BP) is highly active for ORR but tolerant to methanol or CO poisoning, or the adsorption of CO or methanol on single Pt/N-based active sites is weaker than that of $O_2$. The inertness of $Pt_1$-N/BP to both CO and methanol indicates that $Pt_1$-N-based active sites are structurally different from that on traditional Pt-NP-based Pt/C. The difference will be confirmed by the subsequent DFT theoretical results about the weaker adsorption of CO than $O_2$ on $Pt_1$-N-based active sites. Furthermore, based on the US Department of Energy's accelerated durability test protocol, we assessed the durability or long-term operation stability of the $Pt_1$-N/BP catalyst by CV cycling the catalyst between 0.5 and 1.1 $V_{RHE}$ at 200 mV s$^{-1}$ in an $O_2$-saturated 0.1 M HClO$_4$ (refs 28,29). As for the state-of-art commercial Pt/C, as shown in Supplementary Fig. 9, a 34 mV negative shift of $E_{1/2}$ after 10,000 (10 k) CV cycles indicates the deterioration of Pt occurred on Pt/C. While, comparatively, the $Pt_1$-N/BP showed a much better long-term operation stability, as shown in Fig. 2c, the value of $E_{1/2}$ shifted negatively only by 3 mV after 10 k continuous cycles of CV[34]. The high stability of $Pt_1$-N/BP could be attributed to the well-known anchoring effect of doped-N to Pt atoms[18], which has been confirmed by the HAADF images and EXAFS results shown in Fig. 1 and will be further confirmed by the subsequent DFT calculations.

Significantly, the above synergistic-effect-induced high performance (high ORR activity, stability and tolerance to poisoning) of $Pt_1$-N/BP in acid solution was also observed in alkaline condition (0.1 M KOH) as shown in Fig. 2d,e and Supplementary Fig. 10. Compared with pure BP, N/BP or $Pt_1$/BP, the synergistic effect on $Pt_1$-N/BP also can be seen clearly from the much higher $E_{1/2}$ of 0.87 $V_{RHE}$ (blue curve in Fig. 2d), on the same level as the most active Pt-free ORR catalysts in alkaline (Supplementary Table 2)[30,31], and even the traditional state-of-art Pt-NP-based Pt/C with much higher Pt loading (cyan curve in Fig. 2d)[32,33]. As expected, a good durability and tolerance to poisons (CO and methanol) were also observed on $Pt_1$-N/BP in alkaline (Supplementary Fig. 10c and Fig. 2e).

**$H_2$/$O_2$ fuel cell measurement for carbon-supported Pt SAC.** In order to further substantiate the high ORR performance of $Pt_1$-N/BP observed above, we performed acidic $H_2$/$O_2$ fuel cell tests with $Pt_1$-N/BP as cathode catalyst (SM). As shown in Fig. 2f (curves marked with ■), the Nafion-based acidic PEM $H_2$/$O_2$ fuel cell with $Pt_1$-N/BP as cathode (10 μgPt cm$^{-2}$) and commercial Pt/C as anode (80 μgPt cm$^{-2}$) possesses a high performance with maximum power density of 0.68 W cm$^{-2}$ at 80 °C, corresponding to a remarkable Pt utilization efficiency of 0.13 $g_{Pt}$kW$^{-1}$ (Supplementary Table 3). For comparison, acidic PEM $H_2$/$O_2$ fuel cell (curves marked with ★) with commercial Pt/C as both cathode (200 μgPt cm$^{-2}$) and anode (100 μgPt cm$^{-2}$) was also tested at 80 °C. As expected, a much higher maximum power density of 1.02 W cm$^{-2}$ was observed; however, it corresponds to a much

lower Pt utilization efficiency of 0.29 $g_{Pt}$kW$^{-1}$ due to a much higher Pt loading. Moreover, as shown in Fig. 2f, probably due to the mesoporous structure (Supplementary Fig. 11) and the much larger surface area (1,102 m$^2$ g$^{-1}$) of $Pt_1$-N/BP than that of commercial Pt/C, the fuel cell with $Pt_1$-N/BP as cathode at low Pt loading of 10 μgPt cm$^{-2}$ shows performance as high as that with commercial Pt/C as cathode at a much higher Pt loading (200 μgPt cm$^{-2}$) in the current density range < 0.6 A cm$^{-2}$, indicating a much higher Pt utilization of Pt single atom-based $Pt_1$-N/BP than traditional Pt-NP-based Pt/C; however, at higher current density, probably due to significant mass transfer issue in $Pt_1$-N/BP-based thick cathode, the performance of fuel cell with $Pt_1$-N/BP as cathode decays much faster than that with Pt/C as cathode.

Furthermore, the durability of $Pt_1$-N/BP in acidic fuel cell was also evaluated by monitoring the current variation at fixed potential of 0.5 V (refs 35–37). As the insert shows in Fig. 2f, after working 200 h continuously, the current of the fuel cell still remains 74% of the fresh at 80 °C, and 90% remains at 70 °C (Supplementary Fig. 10d), indicating a good durability of $Pt_1$-N/BP as cathode in acidic fuel cells compared with other non-noble ORR catalysts (Supplementary Table 4), consistent with the observations on the three-electrode system (Fig. 2c). All these facts indicate that the Pt single-atom electrocatalyst ($Pt_1$-N/BP) obtained here is indeed one of the most promising alternatives to traditional commercial Pt/C for ORR in fuel cells, whether it be a performance or cost point of view.

**DFT calculations for carbon-supported Pt SAC.** To get more insight into the synergetic effect of single Pt atoms and doped-N or the intrinsic activity of the Pt single atom-based active sites on $Pt_1$-N/BP for ORR, we carried out extensive theoretical investigations using relativistic DFT. Owing to the facts that the majority of doped N on $Pt_1$-N/BP is pyridinic (P)-N (Fig. 1c), and that the P-N has been identified as the strong anchoring point for metal atoms or NPs due to the modified interfacial interaction[9,38–40], here only P-N-based active sites were considered (Supplementary Fig. 12) in the following theoretical calculations. All the electronic structure calculations have been carried out through the spin-polarized DFT calculations as implemented in the Vienna *ab initio* simulation package[41–44]. The formation energies ($E_f$, SM) were calculated to characterize the stability of active sites and the adsorption energies ($E_{ads}$, SM) of adsorbate on different active sites were obtained to identify the thermodynamic stability of the composite systems, a more negative value of $E_{ads}$ signifies greater thermodynamic stability of the composite system. The free energy diagrams of ORR were also calculated according to the method developed by Nørskov *et al.* (SM)[45].

First, we investigated the adsorption of $O_2$ and CO on different $Pt_1$- and P-N-based active sites (Supplementary Figs 12–15), and the corresponding adsorption energy is shown in Supplementary Fig. 16. Interestingly, compared with the well-known stronger adsorption of CO than $O_2$ on Pt NPs (CO poisoning)[46,47], here it shows that only the g-P-N1-$Pt_1$ active site shows inversely stronger $O_2$ adsorption than CO (Supplementary Fig. 16), and the adsorption of CO on g-P-N1-$Pt_1$ is 0.32 eV weaker than that on g-P-N1-$Pt_4$ structure (Supplementary Fig. 17), indicating that the effective sites for ORR on $Pt_1$-N/BP is mainly the P-N1-$Pt_1$-based active sites rather than others since the $Pt_1$-N/BP catalyst has been observed to be tolerant to CO during the ORR process (Fig. 2c).

According to above discussion and the formation energy analysis (Supplementary Table 5), here graphene (g, Fig. 3a) and g-P-N1 (Fig. 3b)-based active sites, g-$Pt_1$ (Fig. 3c) and g-P-N1-$Pt_1$ (Fig. 3d), are considered since previous study has

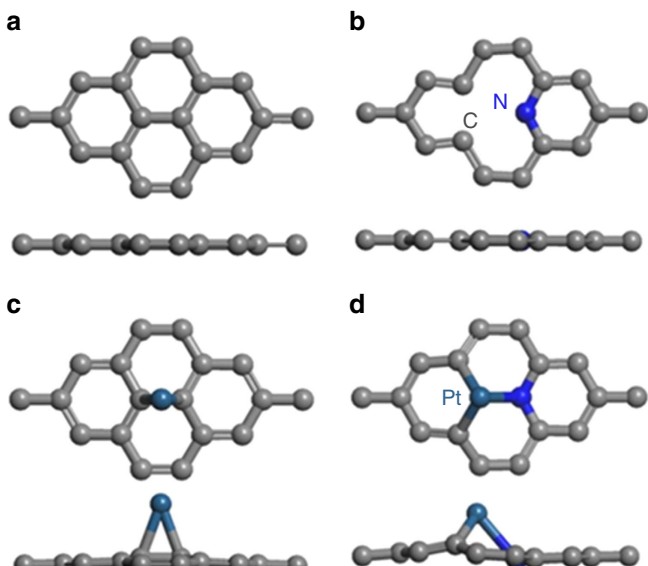

**Figure 3 | Optimized structures of different substrates.** (**a**) Pristine graphene (*g*). (**b**) Pyridinic N1-doped graphene (*g*-P-N1). Optimized structures of (**c**) Pt1 adsorption on pristine graphene (*g*-Pt1), (**d**) Pt1 adsorption on *g*-P-N1 (*g*-P-N1-Pt1). The grey, blue and cyan balls denote the carbon, nitrogen and platinum atoms, respectively.

demonstrated that the defects can be stabilized greatly by the introduction of one P-N (*g*-P-N1; Supplementary Table 5)[9]. The $E_{ads}$ of $Pt_1$ is $-1.56$ eV on pristine graphene and $-5.35$ eV on *g*-P-N1. This suggests that the $Pt_1$ can be selectively trapped strongly by *g*-P-N1 site and thus improve and stabilize the dispersion of Pt single atoms on N-doped graphene surface, confirming the observed good durability of Pt1-N/C in experiments (Fig. 2 and Supplementary Fig. 10d). Furthermore, compared with *g*-Pt1 and *g*-N-Pt1 (Supplementary Fig. 16), the *g*-P-N1-Pt1 site shows much stronger adsorption of $O_2$ with $E_{ads}$ of $-1.93$ eV, indicating that the doped P-N can enhance the $O_2$ adsorption and contributes hugely to the high ORR performance of Pt$_1$-N/C. Therefore, the observed CO tolerance and high ORR activity of the electrocatalyst shown in Fig. 2 can be attributed to the coexistence of both P-N and Pt single atoms in the active site of *g*-P-N1-Pt1. In a word, the above results show that the P-N sites can improve the distribution of Pt single atoms or prevent Pt atoms from aggregating, suppress the CO poisoning and enhance the $O_2$ adsorption on Pt1 sites and finally facilitate the ORR process. All these results are consistent with our experimental observations shown in Fig. 2.

The high ORR performance over *g*-P-N1-Pt1 catalyst is further revealed from the DFT calculations of the catalytic cycle (Fig. 4a). It is found that $O_2$ prefers to bind on the single Pt atom in a side-on configuration with O-O bond length of 1.42 Å (Fig. 4b). It should be noted that the $O_2$ adsorption on bulk Pt is also in a side-on configuration, but the two oxygen atoms adsorb on two or three neighbour Pt atoms, separately, with O-O bond length of 1.37 Å (ref. 48). The longer O-O bond of $O_2$ adsorbed on single Pt atom found here (Fig. 4) probably indicates the easier breaking of O-O bond on *g*-P-N1-Pt1 sites than on traditional bulk Pt-NP-based active sites. The formation of OOH* on *g*-P-N1-Pt1 site continues to elongate the O-O bond length to 1.49 Å, promising an easier dissociation of O-O bond in the subsequent steps. In addition, it is noteworthy that $H_2O_2$ cannot adsorb stably on this substrate since the interaction between $H_2O_2$ and *g*-P-N1-Pt1 directly leads to the decomposition of $H_2O_2$ into two OH groups (Supplementary Fig. 18), which is indicated by the

large O-O bond length of 2.62 Å, supporting a high-efficient 4e reduction process of $O_2$ observed in experiments (Supplementary Fig. 7).

Free energy diagram for ORR on *g*-P-N1-Pt1 site was computed (insert in Fig. 4a, Supplementary Table 6 and Supplementary Table 7) to illuminate the reaction pathways in acidic medium. For comparison, the free energy diagrams for ORR on *g*-P-N1 was also calculated (Supplementary Fig. 19). Since overpotential is an important indicator of the catalytic properties of a catalyst[45,49], thus, here we calculated the ORR overpotential on each catalytic site. From the free energy diagram in Fig. 4, we can learn that the ORR overpotential for *g*-P-N1-Pt1 site is 1.74 V (refs 45,49), and the OH desorption is the rate-limiting step under all potentials. Contrarily, as shown in Supplementary Fig. 19, the reaction pathways of ORR on *g*-P-N1 site are different from that on *g*-P-N1-Pt1 site with a quite large ORR overpotential of 2.87 V, indicating a poorer ORR performance of *g*-P-N1 site compared with that on *g*-P-N1-Pt1 site. While, as for the conventional carbon-supported Pt-NP-based Pt/C, it has been known from both experimental and theoretical points of view that the ORR overpotential on real conventional Pt/C could be $0.69 \sim 1.68$ V (refs 45,50), consistent with our calculation for bulk Pt (111) (1.04 V in Supplementary Fig. 20) and the up limit of 1.68 V is close to the overpotential of 1.74 V on our *g*-P-N1-Pt1 catalyst. Furthermore, as shown in Supplementary Fig. 21 for the ORR reaction pathways on three different active sites, in the kinetic region of the reaction (that is, $U = 0.40$ V), on bulk Pt(111), the $O_2 \rightarrow$ *OOH is an endothermic process, slowing down the whole turnover process; on *g*-P-N1 site, the rate-limiting step (*O→*OH) needs a very high energy of $\sim 2.04$ eV, making the whole ORR process very slow; while on *g*-P-N1-Pt1 site, due to the preceding exothermic steps and the tiny energy barrier (0.91 eV) of the rate-limiting step (the dissociation of *OH), the whole ORR process is much faster than that on *g*-P-N1 site and approximately on the same level as that on bulk Pt(111). These facts unambiguously confirm that the ORR activity of *g*-P-N1-Pt1 catalyst is indeed on the same level as conventional Pt/C due to a synergistic effect between Pt single atoms and P-N. We further studied the ORR process on these three catalysts in an alkaline medium, as shown in Supplementary Figs 22–25 and Supplementary Table 8, which also show the excellent electrocatalytic activity of *g*-P-N1-Pt1.

Moreover, in order to understand the fact that the strong interaction between Pt atoms and the doped-N can hugely prohibit the oxidation of Pt atoms by oxygen in air revealed from the Pt 4f XPS results shown in Fig. 1d,e, we investigated the oxidation resistance of active sites by calculating the formation energies of oxidized *g*-Pt1 and oxidized *g*-P-N1-Pt1 sites. Interestingly, the formation energies are found to be $-2.83$ and $-1.35$ eV for oxidized *g*-Pt1 and oxidized *g*-P-N1-Pt1 sites (Supplementary Fig. 26), respectively. This result indicates that *g*-Pt1 site is much easier to be oxidized in air compared with the *g*-P-N1-Pt1 sites, consistent with the Pt 4f XPS results shown in Fig. 1d,e. Furthermore, the Bader charges are calculated for *g*-P-N1-Pt1, *g*-Pt1 and oxidized *g*-Pt1 (*g*-Pt1-O) systems. Under ideal environment without oxygen ($O_2$) or air, in the first system of *g*-P-N1-Pt1, the charge transfer from Pt1 to *g*-P-N support is 0.222 e and strong charge depletion on Pt1 occurs; while in the second system of *g*-Pt1, Pt1 on pure carbon is negatively charged and possesses $-0.013$ e with a slight electron-accumulation character. These results indicate that the more positively charged Pt1 in *g*-P-N1-Pt1 will lead to better oxidation resistance when exposed to air with oxygen. On the contrary, under oxygen ($O_2$) or air environment the Pt1 atom in the *g*-Pt1 system can be oxidized easily and form PtO by donating the extra electrons to highly electrophilic oxygen due to the electron-accumulation

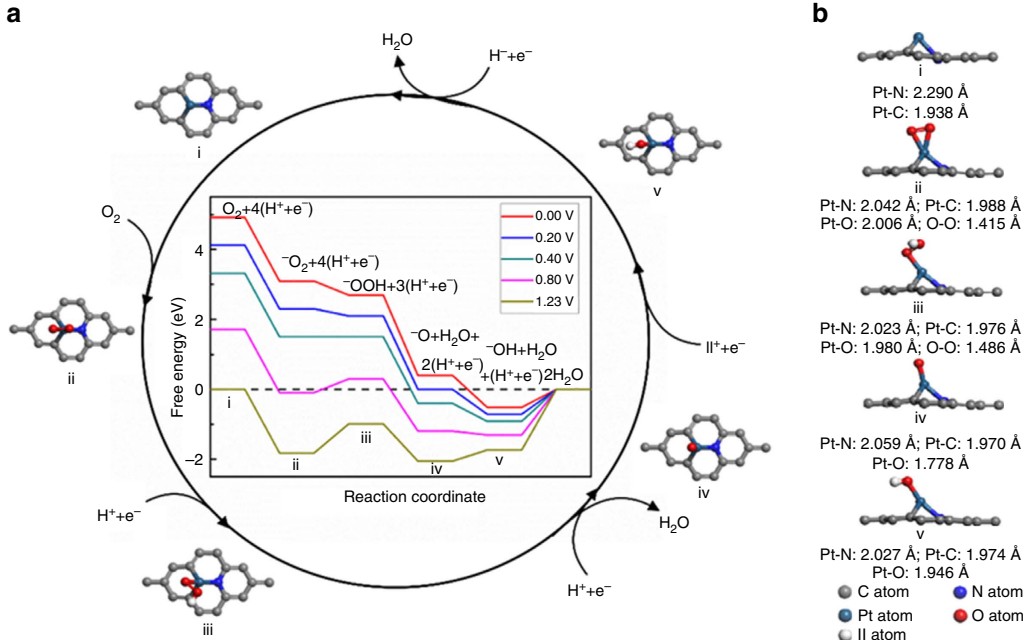

**Figure 4 | The proposed reaction pathways for complete oxygen reduction reaction on the g-P-N1-Pt1 catalyst.** (**a**) Top view, (**b**) side view and bond lengths. The inset in cycle (**a**) shows the free energy diagram for oxygen reduction reaction on the g-P-N1-Pt1 catalyst in acidic medium.

character. Further calculation shows that the Bader charge of Pt1 in oxidized g-Pt1 (g-Pt1-O) system is $+0.420$ e with a remarkable electron-deficient character and more oxidic character compared with that in g-P-N1-Pt1 system, which is consistent with the XPS results shown in Fig. 1d,e. As for the $Pt_1/BP$ without N-doping, in the existent Pt NPs (Fig. 1b), the metal Pt (0) inside particles could be protected from oxidation by the oxidized Pt skin (Supplementary Fig. 27a). That's why the Pt (0) can be detected in $Pt_1/BP$. While as for the $Pt_1$-N/BP, although no Pt NPs were found on it (Fig. 1a), we still cannot exclude the possibility of Pt oxidation on some unfound Pt NPs (Supplementary Fig. 27a); moreover, as shown in Supplementary Fig. 27b, if some individual Pt atoms were far from the C-N centres in distance, then these Pt atoms were the same as the individual Pt atoms on pure carbon and tend to be oxidized by oxygen in air.

Furthermore, as shown in Supplementary Fig. 28, $O_2$ is only adsorbed physically on the oxidized g-Pt1 site, with adsorption energy of $-0.04$ eV and O-O bond length of 1.23 Å, indicating that the $O_2$ molecule is very difficult to be activated on 'oxidized' g-Pt1 site. This result explains very well the inert behaviour of 'oxidized' $Pt_1/BP$ for ORR process observed in experiments (Fig. 2). Therefore, as discussed above, all the theoretical results are consistent with the activity order (Pt-NP/C ≥ $Pt_1$-N/BP > N/BP > oxidized $Pt_1/BP$) or catalytic properties observed in experiments as shown in Fig. 2.

In all, the above results obtained from both experiments and theoretical calculations indicate that Pt SAC-based $Pt_1$-N/BP is a highly efficient and durable electrocatalyst for ORR with the best price-performance ratio ever, making it one of the most promising alternatives to traditional Pt-NP-based electrocatalysts for sustainable, large-scale application of Pt in fuel cells. It should be noted here, as an ORR electrocatalyst for fuel cells, besides its high ORR performance, that the Pt SAC-based $Pt_1$-N/BP overwhelms the reported best Pt-free alternatives for ORR in durability due to the fact that the Pt is much more stable than any other non-Pt atoms in acid; it also overwhelms the traditional Pt-NP-based Pt/C in both tolerance to CO/methanol and Pt utilization efficiency.

## Methods

**Materials.** The carbon black BP2000 was purchased from Asian-Pacific Specialty Chemicals Kuala Lumpur, nitric acid ($HNO_3$) and potassium hydroxide (KOH, $>85.0$) from Beijing Chemical Works, $HClO_4$ (98.0%) and chloroplatinic acid ($H_2PtCl_6 \cdot 6H_2O$, 99.9%) from Beijing Chemical Works, Urea (($NH_2)_2CO$, $>99.0$) from Sinopharm Chemical Reagent Co., LTD, and Nafion solution (5 wt %) were obtained from Sigma-Aldrich. All the chemicals were used as delivered without further treatment. Ultrapure water with the specific resistance of 18.23 MΩ · cm was obtained by reversed osmosis followed by ion-exchange and filtration. RRDE of glassy carbon (4 mm in diameter) was purchased from CH Instruments Inc, USA.

**Catalyst preparation.** In a typical preparation of $Pt_1$-N/BP catalysts, 100 mg of BP2000 and 120 µl of chloroplatinic acid (3.102 $mg_{Pt}$ $ml^{-1}$) were dispersed in 30 ml 6 M $HNO_3$, and refluxed at 80 °C in an oil bath under magnetic stirring for 6 h. The resulting suspension was dried using rotary evaporator at 55 °C and then grounded together with 1.0 g urea. Then, the pyrolysis of the mixed powder was performed at 950 °C for 1 h under argon atmosphere with a flow rate of 80 ml min$^{-1}$. The finally obtained powder was $Pt_1$-N/BP catalysts. For comparison, a pure N-doped carbon (denoted as N/BP) and a pure carbon-supported Pt SAC with Pt 0.4 wt % (denoted as $Pt_1$/BP) were also prepared in a similar way and the BP without Pt and N-doping was also treated in a similar way and denoted as BP. In this work, to obtain the optimal $Pt_1$-N/BP catalysts, the mass ratio of urea and BP was varied in the range of 5–20 and the pyrolysis temperature was varied from 800 to 1,000 °C. During the optimized process, the Pt mass percentages were also varied from 0.1 to 10.

**Catalyst characterization.** The morphology and dimensions of as-prepared samples were obtained using transmission electron microscopy obtained on a JEM-2100F microscopy with an accelerating voltage of 200 kV. Sub-angstrom resolution HAADF-STEM images were obtained on a FEI TITAN Chemi STEM equipped with a CEOS (Heidelburg, Germany) probe corrector, operating at 200 kV. Pt L3-edge absorption spectra (EXAFS) were performed on two beamlines. One was the BL14W1 beamline at the Shanghai Synchrotron Radiation Facility, Shanghai Institute of Applied Physics, China, operated at 3.5 GeV with injection currents of 140–210 mA. In the measurement, a Si(111) double-crystal monochromator was used to reduce the harmonic component of the monochrome beam. Pt foil and $PtO_2$ were used as reference samples and measured in the transmission mode. X-ray diffraction spectrum was obtained from Bruker *D8 ADVANCE* X-ray Diffractometer with using Cu Kα radiation (λ = 0.15418 nm). XPS measurements were performed on a AXIS Ultra DLD (Kratos Company) using a monochromic Al X-ray source.

**Electrochemical measurements.** The activity for the ORR was evaluated by voltamperometry on the NPt-doped carbon material electrodes. Fabrication of the working electrodes was done by pasting catalyst inks on a glassy carbon rotating disk electrode (4 mm in diameter). Its apparent surface area (0.1256 cm$^2$) was used

to normalize the ORR activity of the catalysts. The carbon ink was formed by mixing 5 mg of doped carbon materials catalysts, 50 μl of a 5 wt % Nafion solution in alcohol and 950 μl of ethanol in a plastic vial under ultra-sonication. A 10-μl aliquot of the carbon ink was dropped on the surface of the glassy carbon rotating disk electrode, yielding an approximate catalyst loading of 0.39 mg cm$^{-2}$. For comparison, a commercially available platinum/carbon catalyst, nominally 20 wt % on carbon black from E-TEK, was used. The platinum-based ink was obtained by mixing 1 mg catalyst, 50 μl of a 5 wt % Nafion solution in alcohol and 950 μl of ethanol. Then, a 15 μl aliquot of the platinum ink was dropped on the glassy carbon rotating disk electrode, yielding an approximate loading of 0.12 mg cm$^{-2}$ or 24 μgPt cm$^{-2}$. The electrochemical performance was conducted in 0.1 M HClO$_4$ or 0.1 M KOH solution; the counter and the reference electrodes were a carbon rod and a SCE electrode, respectively. The potential of the electrode was controlled by an EG&G (model 273) potentiostat/galvanostat system. ORR measurements were conducted in oxygen-saturated 0.1 M HClO$_4$ or 0.1 M KOH solution, which was purged with oxygen during the measurement. The scan rate for ORR measurement was 5 mV s$^{-1}$. The ORR polarization curves were collected at 1,600 r.p.m. Long-term operation stability of Pt$_1$-N/BP was performed at room temperature in oxygen-saturated 0.1 M HClO$_4$ or 0.1 M KOH solutions by applying cyclic potential sweeps between 0.5 and 1.1 V versus RHE at a sweep rate of 200 mV s$^{-1}$ for certain amount of cycles. For commercial Pt/C catalyst, the long-term operation stability is evaluated similarly in oxygen-saturated 0.1 M HClO$_4$ solution for 10,000 cycles.

For the calculation of yields of H$_2$O$_2$ on different catalysts, based on both ring and disk currents from RRDE, the percentage of HO$_2^-$ generated from ORR and the electron transfer number ($n$) were estimated by the following equations[51]:

$$HO_2^- \% = 200 \times \frac{i_R/N}{i_D + i_R/N} \quad (1)$$

$$n = 4 \times \frac{i_D}{i_D + i_R/N} \quad (2)$$

Where $i_D$ is the disk current density, $i_R$ is the ring current density and $N$ is the current collection efficiency of the Pt ring disk. $N$ is 0.37 from the reduction of K$_3$Fe [CN]$_6$.

All the current densities have already been normalized to the electrode surface area.

**Single cell tests.** The membrane electrode assembly (MEA) includes cathode gas diffusion layer (GDL), cathode catalyst layer, anode GDL, anode catalyst layer and proton exchange membranes. The GDL on both electrodes was polytetra-fluoroethylene (PTFE)-treated carbon paper (Toray TGP-H-060) covered with 0.4 mg cm$^{-2}$ carbon powder containing 40 wt% PTFE. The cathode catalyst Pt$_1$-N/BP was mixed with Nafion solution (DuPont, 5 wt %) and ethanol with a mass ratio of 1:20:30 to obtain a uniform ink, which was then brushed onto the cathode GDL to obtain the cathode. The loading of Pt$_1$-N/BP on the electrode was 2.5 mg cm$^{-2}$. The anode catalyst layer was prepared by a catalyst-coated-membrane procedure. Specifically, 20 wt% Pt/C (Johnson Matthey), Nafion solution and absolute ethanol with a mass ratio of 1:5:200 were mixed uniformly to obtain catalyst ink, which was directly sprayed on one side of the Nafion212 membrane until the Pt loading is 0.08 mg cm$^{-2}$ to obtain the anode catalyst layer after dried. Then, the MEA components were stacked up in the order of anode GDL, Nafion membrane with the anodic catalyst layer facing down and cathode GDE with the cathodic catalyst layer facing down, and then placed on a hot plate at 130 °C for 120 s without pressure. The obtained MEA was sandwiched between two Au-plating stainless steel bi-polar plate-embedded graphite plates with flow fields. The active area of the MEA is 1 cm$^2$. The single cell was evaluated on a fuel cell test station (Green Light Inc.) at a cell temperature of 80 °C. The anode supply was pure H$_2$ and the flow rate was 100 sccm with 100% humidity. The cathode supply was pure O$_2$ and the flow rate was 200 ml min$^{-1}$ with 100% humidity. The discharging curve was recorded at the back pressure of 0.2 bar. The durability or the lifetime of the single cell was tested at 0.5 V and the back pressure of 0.2 bar, with the other conditions unchanged. The fuel cell with commercial Pt/C as both cathode and anode was prepared and tested in the same way as described above.

**Data availability.** The data that support the findings of this study are available from the authors on reasonable request; see author contributions for specific data sets.

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

## Acknowledgements

Work was funded by the National Basic Research Program of China (973 Program, 2014CB932700), National Natural Science Foundation of China (U1601211, 21633008, 21422307, 21433003, 21503212 and 21503211), the 'Strategic Priority Research Program' of the Chinese Academy of Sciences (XDA09030104), the 'Recruitment Program of Global youth Experts' of China and the Natural Science Foundation of Jilin Province (No. 20130522141JH). The computational resource is partly supported by the Performance Computing Center of Jilin University, China. We are also grateful to the Computing Center of Jilin Province for essential support. D.-J.l. and H.M.B. wish to acknowledge the financial support from US Department of Energy, Office of Energy Efficiency and Renewable Energy, Fuel Cell Technologies Office.

## Author contributions

W.X. conceived and coordinated the research. J.L. contributed to the synthesis of material and the characterization. M.J., P.S., Y.W. and Z.W. contributed to the quantum chemical calculation. L.L and Z.J. contributed to the X-ray absorption fine structure spectroscopy. Y.L., Y.W., H.B., D.L., L.Z., L.J. W.X. and G.S. contributed to the fuel cell measurement. C.M., J.Z., B.Z. and D.S. contributed to the transmission electron microscopy characterization. The manuscript was primarily written by J.L. and W.X. All authors contributed to discussions and manuscript review.

## Additional information

**Competing interests:** The authors declare no competing financial interests.

DOI: 10.1038/ncomms16160    OPEN

# Erratum: High performance platinum single atom electrocatalyst for oxygen reduction reaction

Jing Liu, Menggai Jiao, Lanlu Lu, Heather M. Barkholtz, Yuping Li, Ying Wang, Luhua Jiang, Zhijian Wu, Di-jia Liu, Lin Zhuang, Chao Ma, Jie Zeng, Bingsen Zhang, Dangsheng Su, Ping Song, Wei Xing, Weilin Xu, Ying Wang, Zheng Jiang & Gongquan Sun

Nature Communications 8:15938 doi: 10.1038/ncomms15938 (2017); Published 24 Jul 2017; Updated 27 Sep 2017.

The affiliation details for one of the corresponding authors, Ying Wang, are incorrect in this Article. This author is incorrectly affiliated with 'University of Chinese Academy of Sciences, Beijing 100049, China.' The correct affiliation details for this author are given below:

State Key Laboratory of Rare Earth Resource Utilization, Changchun Institute of Applied Chemistry, Chinese Academy of Sciences, Changchun 130022, China.

