## [Peer Review File · Nature Communications]

Reviewers' comments:

Reviewer #1 (Remarks to the Author):

This work reports high performance in PEMFC using Pt single atom catalyst synthesized with N-doped carbon support. The activity and durability in PEM full cell is astonishing (although the performance in alkaline ORR is not impressive), so it is worthwhile to be published in Nat. Commun. But some of the data were mis-interpreted, thus its correction is required prior to its acceptance.

[1] XPS data in Fig1d,e are weird. Usually, even for Pt SAC, Pt XPS signal is much better. For XPS peak fitting, the peak area ratios for the left and right peak (e.g. Pt4f5/2 vs Pt4f7/2) should be fixed. The fitting in Fig1d,e looks like that four independent peaks were fitted; this should be corrected. Based on this XPS data, authors claimed that Pt on Pt1-N/BP is metallic and Pt on Pt1/BP is oxidic. But this claim is clearly contradictory to the DFT results. DFT calculation showed that Bader charge in Pt1-N/BP is positive and the charge in Pt1/BP is negative. Pt in Pt1-N/BP would have less electrons with electron-deficient character and Pt in Pt1/BP would have more metallic character. Authors kept mentioning the existence of real oxygen on the Pt, but even without oxygen, the electronic structure of Pt in Pt1-N/BP would be very different from Pt in Pt1/BP. Authors should thoroughly correct the discussion about Pt oxidation part.

[2] When the density of active site is low, the mass transfer limitation would play an important role for the cell performance. The information about textural property (surface area and pore volume of Pt1-N/BP) should be provided. In order to compare the mass/electron transfer property, full cell performance of commercial Pt/C in acidic condition should be provided together with Pt1-N/BP.

[3] Real virtue of this paper is the ORR in acidic solution. Pt does not have to be used in alkaline condition. Authors showed the ORR mechanism in alkaline condition in Figure 4, but this should be changed into the mechanism in acidic condition. Furthermore, concentrating the discussion on acidic ORR rather than alkaline ORR would be better to persuade the impact of this paper.

[4] Authors should revise the SI thoroughly. Font style, font size, line spacing, usage of bold character should be edited consistently. In Figure S9, x axis should be the same for (a) and (b). Delta E in (b) should be -19 mV?? There should not be a box in Figure S10(f). ...so many errors/typos.

Reviewer #2 (Remarks to the Author):

The inherently sluggish kinetics of the oxygen reduction reaction (ORR) of Pt at the cathode is the big challenge for widespread commercialization of PEFCs. The single atom catalyst (SAC) is surely of great interest especially for ORR if it can be used in fuel cell techniques. This manuscript reports the preparation of Pt single atoms on nitrogen-doped carbon (N/BP) and their remarkable performance in ORR. The robust ORR performances of SAC in terms of stability and activity are indeed novel. These merits make the manuscript novel enough to be acceptable for publication on Nature Communications. However, the manuscript cannot be published in the current form because of the following issues:

1. In figure 2 and e, why the other samples do not test like Pt/C catalysts, starting from 0 mA/cm². It is suggested that the author normalize all the ORR curve for easy comparison.
2. The authors show the performance of single-cell test for Pt single atom catalyst as cathode in fuel cells. Compared with the results of published papers, Pt1-N/BP catalyst show the best results. But the test conditions vary for different groups. So the authors need to test single-cell performance with commercial Pt/C catalyst as cathode to compare with that of single atoms.
3. Line 215 in page 6 and Supplementary Fig. 10 f, the durability of Pt1-N/BP in acidic fuel cell was also evaluated by monitoring the current variation at fixed potential of 0.5 V. However, the fuel cell do not operate under potential of 0.6 V. The authors need to choose higher potential to

evaluate the durability of single atom catalyst.

4. As we know, the single atoms have selectivity. It was reported that Pt single atom catalysts show very bad or even no ORR activity in acid media (Nat Commun 2016, 7. ; Angew Chem Int Ed Engl 2016, 55 (6), 2058-62.). The authors need to specify why the Pt1-N/BP has good ORR activity and provide the mechanism of Pt1-N/BP for ORR in acid media.

Reviewer #3 (Remarks to the Author):

The authors present a Pt single-atom electrocatalyst for ORR with extensive details on how the catalyst was evaluated and compared with other catalysts and against DOE targets. The identification of individual Pt atoms using HAADF-STEM with atomic resolution supported by other physical characterizations is interesting. However, the claims on the electrochemical performance of the catalysts are somewhat far stretched.

1. It is well known that the biggest benefit of using an alkaline fuel cell system is that Pt/C catalyst is not the most active catalyst as in an acidic system and, therefore, non-noble catalysts are potential choices for an alkaline system. For example, as shown in Fig 2 (a), BP and N/BP (both are non-Pt catalysts) both demonstrated decent performance in KOH solution. So in an alkaline system, Pt loading or Pt utilization is not the best criteria to characterize their Pt single-atom catalyst (Pt-N/BP) against the commercial Pt/C. For instance, the BP and N/BP have zero Pt loading. Based on the results presented in the manuscript, the performance (particularly the durability) of Pt-N/BP catalyst falls under the category of non-noble catalysts despite the presence of very low wt% of Pt. However, the authors emphasized Pt loading (g Pt/kW) in comparing with commercial Pt/C.

2. As far as I know, the 2017&2020 DOE target is 0.125gPt/kW, instead of 0.18 gPt/kW mentioned in the manuscript.

3. The durability test of Pt1-N/BP catalyst and Pt/C should be conducted under the same test conditions, for example, the same number of CV cycles.

4. The DOE 2017 technical target for Pt per kW is required to be obtained at rated power (at $V=0.65$ V) (Citation 39). However, the authors use the peak power density to obtained the Pt loading (gPt/kW). The results can be significantly different!

5. When comparing with Pt single-atom Pt1N/BP, should a Pt1/BP sample with the same Pt loading (prepared with a conventional method) be synthesized and used to emphasize the significant role of single atom Pt? Instead, in the preparation of Pt1/BP, the authors did not use any reducing agents and it is suspected that Pt may not even exist in their Pt1/BP sample at all. This is a serious concern.

6. In spite of the "excellent" performance claimed, the durability data shown in the insert of Figure 2 (f) is nowhere near commercial Pt/C. Again, the performance of the catalyst really falls under the category of "non-noble" catalysts.

7. A Co single-atom electrocatalyst [Co SAs/N-C(900)] was reported lately, which exhibited superior ORR performance with a half-wave potential of 0.881V claiming the best reported non-precious metal catalysts. (Angew. Chem. Int. Ed. 2016, 55, 10800 –10805) . This work may have better implication and should at least be included in Table S1.

Response to the comments

Dear Reviewers,

First of all, Happy new year to all of you.

Also many thanks for your time and work on our manuscript. In the past one month, based on your comments and suggestions, we have revised the manuscript carefully point-by-point by supplementing some new experiments, DFT calculations and discussions. Obviously, your comments and suggestions have helped lot for the improvement of this work. We appreciate it very much. All the details about the revision or the response to the comments can be found in the following red text.

Reviewers' comments:

Reviewer #1 (Remarks to the Author):

This work reports high performance in PEMFC using Pt single atom catalyst synthesized with N-doped carbon support. The activity and durability in PEM full cell is astonishing (although the performance in alkaline ORR is not impressive), so it is worthwhile to be published in Nat. Commun. But some of the data were mis-interpreted, thus its correction is required prior to its acceptance.

[1] XPS data in Fig1d,e are weird. Usually, even for Pt SAC, Pt XPS signal is much better. For XPS peak fitting, the peak area ratios for the left and right peak (e.g. Pt4f5/2 vs Pt4f7/2) should be fixed. The fitting in Fig1d,e looks like that four independent peaks were fitted; this should be corrected. Based on this XPS data, authors claimed that Pt on Pt1-N/BP is metallic and Pt on Pt1/BP is oxidic. But this claim is clearly contradictory to the DFT results. DFT calculation showed that Bader charge in Pt1-N/BP is positive and the charge in Pt1/BP is negative. Pt in Pt1-N/BP would have less electrons with electron-deficient character and Pt in Pt1/BP would have more metallic character. Authors kept mentioning the existence of real oxygen on the Pt, but even without oxygen, the electronic structure of Pt in Pt1-N/BP would be very different from Pt in Pt1/BP. Authors should thoroughly correct the discussion about Pt oxidation part.

Response to the comment: Dear reviewer, thanks for your comment here. Based on it, we have done the refitting of the Pt XPS signal with fixed ratios of the left and right peaks (John F. Moulder, William F. Stickle, Peter E. Sobol, Kenneth D. Bomben, in *Handbook of X Ray Photoelectron Spectroscopy*. J. Chastain, Editor, Perkin Elmer Corp., Eden Prairie, MN (1992)). Corresponding revision also have been done. More details can be found in the revised manuscript (around Fig. 1d,e). Thanks for your reminder here.

Sorry for the confusing on the XPS and DFT Bader charge discussions. As shown in Fig. 1d, e, it can be seen clearly that the obtained real catalyst of Pt1-N/BP contains more Pt(0) and less Pt(II) than the real Pt1/BP, indicating it is easier for Pt1/BP to be oxidized than Pt1-N/BP. In order to understand the properties of Pt single atom on different substrates, DFT calculations were carried out based on two ideal models g-P-N1-Pt1 and g-Pt1. The DFT calculations show, under ideal environment without oxygen (O₂) or air, the Pt1 atom possesses +0.222 e in g-P-N1-Pt1 system, while it (Pt1) is negatively charged by -0.013 e in g-Pt1 system. Therefore, the Pt1 in g-Pt1 system tends to be oxidized and form PtO by donating extra electrons to oxygen when exposure to air, the "oxidized" g-Pt1 corresponds to the real catalyst Pt1/BP with high content of Pt(II) or PtO (Fig. 1e); while the Pt1 in g-P-N1-Pt1 system can resist the oxidation when exposure to air due to the electron-depletion character and can be stable in non-oxidized state of Pt(0) (Fig. 1d). That means the two real catalysts Pt1/BP and Pt1-N/BP we finally obtained from experiments correspond to the "oxidized" g-Pt1 (PtO) and "non-oxidized" g-P-N1-Pt1 Pt(0), respectively. That's why our XPS results (Fig. 1d,e) show that the Pt1-N/BP contains more Pt(0) and less Pt(II) than Pt1/BP.

Thanks for your reminder here. Based on it, in order to avoid any confusing, we have done the following revision in the manuscript (Page-3) from the old version "most of the Pt atoms (86.6%) on it are metallic, only a small amount of them were oxidized as PtO " to " most of the Pt atoms

(86.6%) on it are metal Pt(0), only a small amount of them were oxidized as PtO " .

[2] When the density of active site is low, the mass transfer limitation would play an important role for the cell performance. The information about textural property (surface area and pore volume of Pt1-N/BP) should be provided. In order to compare the mass/electron transfer property, full cell performance of commercial Pt/C in acidic condition should be provided together with Pt1-N/BP.

Response to the comment: Thanks for your comment here. Based on it, we have supplemented the BET and pore size distribution analysis about the Pt1-N/BP. More discussion can be found in the revised manuscript (Page-6) and SI (Supplementary Fig. 11). Furthermore, based on your comment, we have supplemented the fuel cell test with commercial Pt/C as cathode in acidic condition. More details can be found in the revised Fig. 2f.

[3] Real virtue of this paper is the ORR in acidic solution. Pt does not have to be used in alkaline condition. Authors showed the ORR mechanism in alkaline condition in Figure 4, but this should be changed into the mechanism in acidic condition. Furthermore, concentrating the discussion on acidic ORR rather than alkaline ORR would be better to persuade the impact of this paper.

Response to the comment: Thanks for your suggestion. Based on it, we have revised the manuscript by concentrating the discussion on acidic ORR. More details can be found in the revised manuscript around Fig. 2.

Moreover, based on your comment, Figure 4 was also replaced with the ORR mechanism in acidic condition. The corresponding discussion in the main text was also revised to concentrate on the ORR process in acidic condition, as follows (Page-9 in the revised manuscript):

“Free energy diagram for oxygen reduction reaction on g-P-N1-Pt1 site was computed (insert in Fig. 4a, Supplementary Table 5, and Supplementary Table 6) to illuminate the reaction pathways in acidic medium. For comparison, the free energy diagrams for ORR on g-P-N1 was also calculated (Supplementary Fig. 19). Since overpotential is an important indicator of the catalytic properties of a catalyst^{46,50}, thus, here we calculated the ORR overpotential on each catalytic site. From the free energy diagram in Fig. 4, we can learn that the ORR overpotential for g-P-N1-Pt1 site is 1.74 V^{46,50}, and the OH desorption is the rate-limiting step under all potentials. Contrarily, as shown in Supplementary Fig. 19, the reaction pathways of ORR on g-P-N1 site are different from that on g-P-N1-Pt1 site with a quite large ORR overpotential of 2.87 V, indicating a poorer ORR performance of g-P-N1 site compared with that on g-P-N1-Pt1 site. While, as for the conventional carbon-supported Pt-NP-based Pt/C, it has been known from both experimental and theoretical points of view that the ORR overpotential on real conventional Pt/C could be 0.69~1.68 V^{46,51}, consistent with our calculation for bulk Pt (111) (1.04 V in Supplementary Fig. 20) and the up limit of 1.68 V is close to the overpotential of 1.74 V on our g-P-N1-Pt1 catalyst. Furthermore, as shown in Supplementary Fig. 21 for the ORR reaction pathways on three different active sites, in the kinetic region of the reaction (i.e., U=0.40 V), on bulk Pt(111), the $O_2 \rightarrow *OOH$ is an endothermic process, slowing down the whole turnover process; on g-P-N1 site, the rate-limiting step ($*O \rightarrow *OH$) needs a very high energy of about 2.04eV, making the whole ORR process relatively very slow; while on g-P-N1-Pt1 site, due to the preceding exothermic steps and the tiny energy barrier (0.91 eV) of the rate-limiting step (the dissociation of $*OH$), the whole ORR process is much faster than that on g-P-N1 site and approximately on the same level as that on bulk Pt(111). These facts unambiguously confirm that the ORR activity of g-P-N1-Pt1 catalyst is indeed on the same level as

conventional Pt/C due to a synergistic effect between Pt single atoms and P-N. We further studied the ORR process on these three catalysts in alkaline medium, as shown in Supplementary Fig. 22-25 and Table S7, which also show the excellent electrocatalytic activity of g-P-N1-Pt1.”

[4] Authors should revise the SI thoroughly. Font style, font size, line spacing, usage of bold character should be edited consistently. In Figure S9, x axis should be the same for (a) and (b). Delta E in (b) should be -19 mV?? There should not be a box in Figure S10(f). ...so many errors/typos.

Response to the comment: Thanks for your reminder here. Based on it, we have done all the corresponding revision thoroughly.

Reviewer #2 (Remarks to the Author):

The inherently sluggish kinetics of the oxygen reduction reaction (ORR) of Pt at the cathode is the big challenge for widespread commercialization of PEFCs. The single atom catalyst (SAC) is surely of great interest especially for ORR if it can be used in fuel cell techniques. This manuscript reports the preparation of Pt single atoms on nitrogen-doped carbon (N/BP) and their remarkable performance in ORR. The robust ORR performances of SAC in terms of stability and activity are indeed novel. These merits make the manuscript novel enough to be acceptable for publication on Nature Communications. However, the manuscript cannot be published in the current form because of the following issues:

1. In figure 2a and e, why the other samples do not test like Pt/C catalysts, starting from 0 mA/cm². It is suggested that the author normalize all the ORR curve for easy comparison.

Response to the comment: Thanks for your comment here. Actually, the data shown in Fig. 2a,2e are the original data without normalization. Now in the revised manuscript, based on your comment, we normalized all the samples to start from 0 mA/cm² in Fig.2 for easy comparison.

2. The authors show the performance of single-cell test for Pt single atom catalyst as cathode in fuel cells. Compared with the results of published papers, Pt1-N/BP catalyst show the best results. But the test conditions vary for different groups. So the authors need to test single-cell performance with commercial Pt/C catalyst as cathode to compare with that of single atoms.

Response to the comment: Thanks for your comment here. Based on it, we have supplemented the fuel cell test with commercial Pt/C as cathode for comparison. More details can be found in the revised manuscript around Fig. 2f.

3. Line 215 in page 6 and Supplementary Fig. 10 f, the durability of Pt1-N/BP in acidic fuel cell was also evaluated by monitoring the current variation at fixed potential of 0.5 V. However, the fuel cell do not operate under potential of 0.6 V. The authors need to choose higher potential to evaluate the durability of single atom catalyst.

Response to the comment: Thanks for your comment here. Actually, in previous references, the fixed potential of 0.5 V was extensively adopted to evaluate the durability of catalysts in acidic fuel cells (such as *Science* **2009**, 324, 71; *Angew. Chem.* **2015**, 127, 10045; *PNAS*, 2015, 112, 10629). For a better comparison of our catalyst with these previous ones, the same fixed potential of 0.5 V was adopted in this work. To make it clear, we have supplemented a few relevant references to the revised manuscript (Page-6, Ref. 36, Ref.37 and Ref.38).

4. As we know, the single atoms have selectivity. It was reported that Pt single atom catalysts show

very bad or even no ORR activity in acid media (Nat Commun 2016, 7. ; Angew Chem Int Ed Engl 2016, 55 (6), 2058-62.). The authors need to specify why the Pt1-N/BP has good ORR activity and provide the mechanism of Pt1-N/BP for ORR in acid media.

Response to the comment: Thanks for your comment here. As for the good ORR performance of Pt1-N/BP, the experimental data shown in Fig. 2 indicates the high ORR performance could be due to the synergistic effect between doped N and Pt single atoms. Such synergistic effect was further confirmed by the DFT results shown in the manuscript (around Fig. 4 and Supplementary Figs.12-28). Based on your comment, to understand such effect clearly or specify why Pt1-N/BP has good ORR activity in acid media, the ORR mechanism on g-P-N1-Pt1 in alkaline condition (Figure 4 in the old manuscript) has been changed to the ORR mechanism in acid. Also, the corresponding discussion in the main text is revised accordingly (Page-9,10).

Reviewer #3 (Remarks to the Author):

The authors present a Pt single-atom electrocatalyst for ORR with extensive details on how the catalyst was evaluated and compared with other catalysts and against DOE targets. The identification of individual Pt atoms using HAADF-STEM with atomic resolution supported by other physical characterizations is interesting. However, the claims on the electrochemical performance of the catalysts are somewhat far stretched.

1. It is well known that the biggest benefit of using an alkaline fuel cell system is that Pt/C catalyst is not the most active catalyst as in an acidic system and, therefore, non-noble catalysts are potential choices for an alkaline system. For example, as shown in Fig 2 (a), BP and N/BP (both are non-Pt catalysts) both demonstrated decent performance in KOH solution. So in an alkaline system, Pt loading or Pt utilization is not the best criteria to characterize their Pt single-atom catalyst (Pt-N/BP) against the commercial Pt/C. For instance, the BP and N/BP have zero Pt loading. Based on the results presented in the manuscript, the performance (particularly the durability) of Pt-N/BP catalyst falls under the category of non-noble catalysts despite the presence of very low wt% of Pt. However, the authors emphasized Pt loading (g Pt/kW) in comparing with commercial Pt/C.

Response to the comment: Thanks for your comment here. Based on it, we have revised the discussion about Pt loading or utilization in the manuscript. Also it should be noted here, the parameter of Pt loading or consumption in fuel cell (g Pt/kW) adopted here is based on the total Pt loading or consumption in a whole single cell, including both cathode and anode. Just like you said, the low or zero Pt consumption in one side does not mean much. So, the Table S3 is based on such parameter (g Pt/kW) from the total Pt loading or consumption (both cathode and anode) in single acidic H₂/O₂ fuel cell always with traditional Pt/C as anode. Moreover, in the revised manuscript, based on the comments from both you and the first reviewer, we have refocused the work on the performance in acidic condition. The Fig. 2, Fig.4 and several Supplementary figures in SI were replaced accordingly with acidic results. More details can be found in the revised manuscript.

2. As far as I know, the 2017&2020 DOE target is 0.125gPt/kW, instead of 0.18 gPt/kW mentioned in the manuscript. The DOE 2017 technical target for Pt per kW is required to be obtained at rated power (at V=0.65 V) (Citation 39). However, the authors use the peak power density to obtain the Pt loading (gPt/kW). The results can be significantly different!

Response to the comment: Thanks for your reminder here. Based on it, in order to avoid any

confusing or misunderstanding, we have removed the statement about the comparison between our result and DOE target in the revised manuscript (Page-6).

3. The durability test of Pt₁-N/BP catalyst and Pt/C should be conducted under the same test conditions, for example, the same number of CV cycles.

Response to the comment: Thanks for your reminder here. Based on it, we supplemented new experiments to compare the durability of these two catalysts (Fig. 2d vs. Supplementary Fig. 9) after the same number (10k) of CV cycles in acid.

4. When comparing with Pt single-atom Pt₁N/BP, should a Pt₁/BP sample with the same Pt loading (prepared with a conventional method) be synthesized and used to emphasize the significant role of single atom Pt? Instead, in the preparation of Pt₁/BP, the authors did not use any reducing agents and it is suspected that Pt may not even exist in their Pt₁/BP sample at all. This is a serious concern.

Response to the comment: Thanks for your comment here. Actually, both Pt₁-N/BP and Pt₁/BP possess the same Pt loading confirmed by ICP analysis. For the preparation of these two catalysts, the salt of Pt was pyrolyzed to metal Pt at 950 °C under argon atmosphere with flow rate of 80 mL/min. No reducing agents are needed in such pyrolyzing method (*Catal. Lett.*, **1887**, 33, 405-410; *Chem. Mater.*, **2005**, 17, 6624–6634). More details can be found in the SI about the preparation of different catalysts.

5. In spite of the “excellent” performance claimed, the durability data shown in the insert of Figure 2 (f) is nowhere near commercial Pt/C. Again, the performance of the catalyst really falls under the category of “non-noble” catalysts.

Response to the comment: Thanks for your comment here. Now, in the revised manuscript, we only compared our catalyst with some other reported “non-noble” catalysts (Table S1, Table S2 and Table S3) in the manuscript.

6. A Co single-atom electrocatalyst [Co SAs/N-C(900)] was reported lately, which exhibited superior ORR performance with a half-wave potential of 0.881V claiming the best reported non-precious metal catalysts. (*Angew. Chem. Int. Ed.* 2016, 55, 10800 –10805) . This work may have better implication and should at least be included in Table S1.

Response to the comment: Thanks for your reminder. Based on it, we have supplemented this work to the Table S2 for the performance comparison in alkaline condition.

Reviewers' comments:

Reviewer #1 (Remarks to the Author):

The concept that a pair of Pt single atom-N embedded in carbon can actually catalyze 4e- ORR is interesting. However, the authors' strongest claim is that this catalyst system can be an actual alternative of current Pt/C catalyst system used in PEMFC with minimized Pt utilization, and I think that the claim was not supported by the experimental data properly.

- [1] Figure 2f (comparison of Pt1-N and Pt/C) clearly showed that Pt1-N is not as good as commercial Pt/C mainly due to mass-transfer limitation, although the catalytic activity is better in Pt1-N (which is contradictory to the half cell results). The discussion in line 200~205 is wrong.
- [2] In half cell test (Fig 2a), Pt/C should have a diffusion-limiting current at 5.7~5.8 mA/cm², but this work showed smaller value of ~5.3. I doubt if the electrode is prepared in a correct way.
- [3] Comparing diffusion-limiting current at 0.2V is non-sense (Fig 2b). Mass activity for ORR is usually compared at 0.9 V using a proper equation.
- [4] The comparison in Table S3 (supplementary information) is meaningless. For non-Pt catalysts used in cathode, people often use an excessive amount of Pt catalyst to minimize any limitation in anode part, and that is why people use rather excessive amounts of Pt in anode. In Table S3, other works used 0.25-0.4 mgPt/cm², but practically, people usually use only 0.1 or 0.2 mgPt/cm² in the anode. Here, authors claimed that 0.09 gPt/kW, but in Pt/C the number was only slightly higher as 0.10 gPt/kW. People can make a lot higher power density using Pt/C, comparing to this work. Also authors used iR-free cell voltage, but this comparison in full cell test is rather weird. I doubt that other works in Table S3 would not use this 'iR' correction.
- [5] The durability of Pt1-N (inset of Fig 2f) is not so good. Even, non-Pt carbon-based catalyst can have better durability than this.
- [6] I still think that discussion about Bader charge (line 311~318) is wrong. Pt on Pt1-N is +0.222 and Pt on g-Pt1 is -0.013, this is clearly contradictory to XPS results. This contradiction would recommend that the DFT model is not consistent with the real sample.

Reviewer #2 (Remarks to the Author):

The authors have significantly improved the original manuscript and most of my questions have been well addressed. I am happy to recommend the acceptance of this manuscript for publication in Nature Communications after the authors consider the following minor points:
The method of MEA fabrication is very important. Why use different methods to make anode catalyst layer and cathode catalyst layer? It is suggested that the author provide more detail about how make ink, like the ration of catalyt , Nafion and ethanol.

Reviewer #3 (Remarks to the Author):

The changes made to the original manuscripts should be highlighted. It is otherwise (as it is now without highlighting) very difficult for the reviewer to dig into where the changes have been made in response to the comments.

Response to the comments

Many thanks for your time and work on our manuscript. Based on the comments and suggestions from reviewers, we have revised the manuscript point-by-point carefully. Obviously, these comments and suggestions have helped lot for the further improvement of this work. We appreciate your kind help very much. All the details about the revision or the response to the comments can be found in the following red text or in the revised manuscript highlighted in red.

Reviewers' comments:

Reviewer #1 (Remarks to the Author):

The concept that a pair of Pt single atom-N embedded in carbon can actually catalyze 4e- ORR is interesting. However, the authors' strongest claim is that this catalyst system can be an actual alternative of current Pt/C catalyst system used in PEMFC with minimized Pt utilization, and I think that the claim was not supported by the experimental data properly.

[1] Figure 2f (comparison of Pt1-N and Pt/C) clearly showed that Pt1-N is not as good as commercial Pt/C mainly due to mass-transfer limitation, although the catalytic acidity is better in Pt1-N (which is contradictory to the half cell results). The discussion in line 200~205 is wrong.

Response to the comment: Dear Editor, thanks for your comment. Based on it, we have reconsidered our previous discussion you mentioned and realized it is indeed not so appropriate. Now in the revised manuscript, based on your comments, we have revised the discussion you mentioned (line 200~205) (Page-6 highlighted with red) as the following: "Moreover, as shown in Fig. 2f, probably due to the mesoporous structure (Supplementary Fig. 11), the much larger surface area ($1102 \text{ m}^2 \text{ g}^{-1}$) and higher Pt utilization (Fig. 2b) of Pt₁-N/BP than that of Pt/C, the fuel cell with Pt₁-N/BP as cathode shows a better performance than that with Pt/C as cathode in the current density range $< 1.5 \text{ A cm}^{-2}$; while at higher current density or in the mass transport region, the performance of fuel cell with Pt/C as cathode exceeds that with Pt₁-N/BP as cathode. Such observation at high current density probably could be attributed to the mass transfer limitation in the Pt₁-N/BP-based cathode due to the much thicker catalyst layer used ($2.5 \text{ mg}_{\text{Pt1-N/BP}} \text{ cm}^{-2}$) than that of Pt/C-based ($0.2 \text{ mg}_{\text{Pt/C}} \text{ cm}^{-2}$) cathode."

Indeed, the fuel cell data showed that Pt1-N is not as good as commercial Pt/C at high current density mainly due to mass-transfer limitation. While in the lower current density region, the fuel cell performance with Pt1-N as cathode is better than that with Pt/C as cathode. This result is apparently different from that shown in the half cell, which shows that the apparent ORR performance of Pt1-N is lower than that of commercial Pt/C. While such difference could be due to the following reasons: in the half cell test (Fig. 2a), the Pt loading (24 ug Pt cm^{-2}) of Pt/C is more than 10 times higher than that ($1.56 \text{ ug Pt cm}^{-2}$) of Pt1-N; while in the fuel cell tests (Fig. 2f), the Pt loading ($0.04 \text{ mg}_{\text{Pt}} \text{ cm}^{-2}$) in Pt/C cathode is 4 times of the Pt loading ($0.01 \text{ mg}_{\text{Pt}} \text{ cm}^{-2}$) in Pt1-N cathode. Due to the higher utilization efficiency of Pt in Pt1-N than that in Pt/C (Fig. 2b), it is possible for us to observe higher fuel cell performance with Pt1-N as cathode. Moreover, the fuel cell performance usually systematically reflects a complicated interaction among multiple factors (for instance, the different types/amounts of catalysts or the thickness of the catalyst layer can lead to different properties of triple phase boundary of MEA, which can hugely affect the performance of fuel cell); while the half cell test usually only can simply reflect the activity of a catalyst in aqueous environment.

[2] In half cell test (Fig 2a), Pt/C should have a diffusion-limiting current at 5.7~5.8 mA/cm², but this work showed smaller value of ~5.3. I doubt if the electrode is prepared in a correct way.

Response to the comment: Thanks for your comment here. Based on it, we read a few references related to it and realized, just like you said, theoretically, the diffusion-limiting current of ORR on Pt electrode is about 5.7 mA cm⁻² (*Electrochim. Acta* **2008**, 53, 3181–3188; *Anal. Chem.* **2010**, 82, 6321–6328). But in reality, due to effect of some unavoidable factors in real experiment, the measured limiting current of ORR on Pt is usually not exactly the same as the theoretical value. It has been reported that the value of limiting current obtained from experiment could be reliable if it is within the 10% margin of the theoretical value (*Anal. Chem.* **2010**, 82, 6321–6328). For the case here, our value of 5.3 is within the 7% (< 10%) margin of the theoretical value of 5.7. So we can tell that the method for the preparation of electrode in our lab is indeed correct or reliable. So dear reviewer, thank you again for your comment here. We learnt lot from it.

[3] Comparing diffusion-limiting current at 0.2V is non-sense (Fig 2b). Mass activity for ORR is usually compared at 0.9 V using a proper equation.

Response to the comment: Thanks for your comment. Based on it, we revised the manuscript with a simple current comparison between these two catalysts at 0.9 V. More details can be found in the revised manuscript (Page-5 highlighted with red).

[4] The comparison in Table S3 (supplementary information) is meaningless. For non-Pt catalysts used in cathode, people often use an excessive amount of Pt catalyst to minimized any limitation in anode part, and that is why people use rather excessive amounts of Pt in anode. In Table S3, other works used 0.25-0.4 mgPt/cm², but practically, people usually use only 0.1 or 0.2 mgPt/cm² in the anode. Here, authors claimed that 0.09 gPt/kW, but in Pt/C the number was only slightly higher as 0.10 gPt/kW. People can make a lot higher power density using Pt/C, comparing to this work. Also authors used *iR*-free cell voltage, but this comparison in full cell test is rather weird. I doubt that other works in Table S3 would not use this '*iR*' correction.

Response to the comment: Thanks for your comment here. Indeed, in previous fuel cells with non-Pt cathode catalysts, just like you said, "people often use an excessive amount of Pt catalyst to minimized any limitation in anode part, and that is why people use rather excessive amounts of Pt in anode ". Since you believe that the comparison in Table S3 is meaningless, then we removed the corresponding discussion about Table S3, but the comparison of Pt utilization efficiency (in g_{Pt}/kW) between Pt1-N and commercial Pt/C in fuel cells was kept since no excessive amount of Pt was used in the anodes of these two cases. As for the power density of Pt/C, just like you said, people can make a lot higher/lower power density with higher/lower Pt loading. As for the *iR*-free cell voltages (*R* is the ohmic resistance of the fuel cell for this case), we adopted these values just because we wanted to evaluate the maximum performance of catalyst in fuel cell by removing the effect from the ohmic resistance (*R*) of fuel cells since it has been known that the *R* is mainly from the polymer electrolyte membrane. But, since you commented that it is inappropriate, then based on your opinion we replaced the old fuel cell data with the ones without *iR* correction. It can be seen that the same conclusion or discussion can be made from the new data (revised Fig. 2f and Page-6 highlighted with red in the revised manuscript).

[5] The durability of Pt1-N (inset of Fig 2f) is not so good. Even, non-Pt carbon-based catalyst can have better durability than this.

Response to the comment: We agree with you on that even some " non-Pt carbon-based catalyst can have better durability than this ", but to our best knowledge, the observed durability of this catalyst in fuel cell (inset of Fig 2f, 74% of current remained after 200 hours at 0.5 V) is indeed better than that of many recently reported non-Pt carbon-based catalysts (such as the one (only 28% of current remained after 100 hours at 0.5 V) reported by *Angew. Chem. Int. Ed.* 2015, 54, 9907).

[6] I still think that discussion about Bader charge (line 311~318) is wrong. Pt on Pt1-N is +0.222 and Pt on g-Pt1 is -0.013, this is clearly contradictory to XPS results. This contradiction would recommend that the DFT model is not consistent with the real sample.

Response to the comment: Thanks for your comment. To further make it clear to you, it should be noted here that the results, the bader charge of Pt on g-P-N1-Pt1 is +0.222 and the bader charge of Pt on g-Pt1 is -0.013, are obtained from calculation under ideal environment without oxygen (O₂) or air. Based on these results we would like to clarify that the former positively charged Pt (+0.222) possesses electron-depletion character and then is difficult to be further oxidized to form PtO by oxygen when exposed to O₂ environment in the later process. On the contrary, the latter negatively charged Pt (-0.013) on g-Pt1 possesses an electron-accumulation character, which will make Pt1 easy to be oxidized (to form PtO) by donating the extra electrons to highly electrophilic oxygen. This difference between g-P-N1-Pt1 and g-Pt1 is actually supported by the calculated formation energies of oxidized g-Pt1 and oxidized g-P-N1-Pt1 sites, more details can be found in the manuscript (Page-10 highlighted with blue). In the present revision, to further clarify it, we supplemented the calculation of the Bader charge of Pt1 in oxidized g-Pt1 (g-Pt1-O). Interestingly, a more positively charged character (+0.420 e) of Pt1 in g-Pt1-O than Pt1(+0.222 e) in g-P-N1-Pt1 was obtained, which is consistent with the XPS results (Fig.1d,e).

Based on above supplemented new calculation, the previous discussion (Page-10 in the revised manuscript) has been revised from "Furthermore, the bader charge is calculated for both g-P-N1-Pt1 and g-Pt1 systems. In the former system, the charge transfer from Pt1 to g-P-N1 support is 0.222 e and strong charge depletion on Pt1 occurs. Whereas, Pt1 on pure carbon is negatively charged and possesses -0.013 e in the g-Pt1 system. Thus, Pt1 in g-P-N1-Pt1 will be more positively charged compared to that in Pt-C system, which then leads to better oxidation resistance of g-P-N1-Pt1 site" to "Furthermore, the Bader charges are calculated for g-P-N1-Pt1, g-Pt1 and oxidized g-Pt1 (g-Pt1-O) systems. Under ideal environment without oxygen (O₂) or air, in the first system of g-P-N1-Pt1, the charge transfer from Pt1 to g-P-N1 support is 0.222 e and strong charge depletion on Pt1 occurs; whereas, in the second system of g-Pt1, Pt1 on pure carbon is negatively charged and possesses -0.013 e with a slight electron-accumulation character. These results indicate that the more positively charged Pt1 in g-P-N1-Pt1 will lead to better oxidation resistance when exposed to air with oxygen. On the contrary, under oxygen (O₂) or air environment the Pt1 atom in g-Pt1 system can be oxidized easily and form PtO by donating the extra electrons to highly electrophilic oxygen due to the electron-accumulation character. Further calculation shows that the Bader charge of Pt1 in oxidized g-Pt1 (g-Pt1-O) system is +0.420 e with a

remarkable electron-deficient character and more oxidic character compared with that in g-P-NI-Pt1 system, which is consistent with the XPS results shown in Fig.1d, e.”

Reviewer #2 (Remarks to the Author):

The authors have significantly improved the original manuscript and most of my questions have been well addressed. I am happy to recommend the acceptance of this manuscript for publication in Nature Communications after the authors consider the following minor points:

The method of MEA fabrication is very important. Why use different methods to make anode catalyst layer and cathode catalyst layer? It is suggested that the author provide more detail about how make ink, like the ration of catalyst , Nafion and ethanol.

Response to the comment: Dear Reviewer, many thanks for your comment and suggestion. Indeed, the method of MEA preparation is very important. In fuel cell study, in order to maximize the performance of a catalyst in fuel cell, the optimal method for the preparation of catalyst layer could be different among different catalysts. For the case here, the fabrication methods of both electrodes were varied from the different catalyst loadings, i.e., for the cathode of Pt₁-N/BP, the catalyst loading is as high as 2.5 mg cm⁻², so the brushing method was employed to avoid cracking of the thick catalyst layer, while for the anode, the catalyst loading is as low as 0.08 mg cm⁻², thus the spraying method was employed to ensure a close contact with the Nafion membrane. The detailed description (the following paragraph in blue) for the preparation of catalyst inks has been supplemented to the revised Supplementary Information (Page-S4) with more details, including the ratios of the catalyst, Nafion and ethanol.

" The cathode catalyst Pt₁-N/BP was mixed with Nafion solution (DuPont, 5 wt %) and ethanol with a mass ratio of 1:20:30 to obtain a uniform ink, which was then brushed onto the cathode GDL to obtain the cathode. The loading of Pt₁-N/BP on the electrode was 2.5 mg cm⁻². The anode catalyst layer was prepared by a catalyst-coated-membrane procedure. Specifically, 20 wt% Pt/C (Johnson Matthey), Nafion solution and absolute ethanol with a mass ratio of 1:5:200 were mixed uniformly to obtain catalyst ink, which was directly sprayed on one side of the Nafion212 membrane until the Pt loading is 0.08 mg cm⁻² to obtain the anode catalyst layer after dried. "

Reviewer #3 (Remarks to the Author):

The changes made to the original manuscripts should be highlighted. It is otherwise (as it is now without highlighting) very difficult for the reviewer to dig into where the changes have been made in response to the comments.

Response to the comment: Dear reviewer, thanks for your suggestion here. Also we are very sorry for the inconvenience. Actually, in our last revision, a Word file with changes highlighted in red has been uploaded as "Related Materials for review only" along with the manuscript file without highlight. In order to avoid such inconvenience, in the present revision, we just include all the revision/changes highlighted in red in the main text file for submission so that you can see them directly.

In all, dear reviewers, many thanks for your time and work on our manuscript. Your comments and

suggestions have helped lot for the further improvement of this work. We appreciate it very much!

Reviewers' comments:

Reviewer #1 (Remarks to the Author):

The authors addressed most of my questions properly. But I still think that durability of Pt1-N is not so good, so I recommend to use 'good durability' in the manuscript instead of 'remarkable durability' (line 211).

Reviewer #2 (Remarks to the Author):

The authors have addressed well all suggestions and comments. I think that it is publishable now.

Reviewer #3 (Remarks to the Author):

1. I still think the claim of "super low Pt loading" and "high Pt utilization are misleading as I pointed out in my first Review that the ORR catalyst in this manuscript essentially belongs to a non-noble catalyst and as a result Pt utilization is not an appropriate characterization. For example, Fig 2 b's comparison is very misleading between Pt/C and a non-noble catalyst.

2. Another more misleading comparison is single cell performance in acid described in lines 191 and 207 referred to Fig 2f.. Everyone in this field knows that the state-of-the art commercial Pt/C is optimally employed for commercial CCMs (such as Ballard CCMs and Gore CCMs) with Pt loading ranging between 0.15 and 0.4 mgPt/cm² for the cathode side. Now the author manufactured a CCM with Pt/C commercial with only 0.04 mgPt/cm² and tested the cell performance with H₂/O₂ instead of H₂/air. I believe the author understands this loading and the H₂/O₂ are the best condition for his/her non-noble type catalyst and the non-noble type catalyst would be running into significant mass transfer issue due to the large thickness if the CCM were made with higher Pt loading and tested in H₂/air system.

3. The durability result shown in the insert of Fig 2f is really not satisfying. People have shown the durability of commercial Pt/C in thousands of hours (before the voltage drops to 80% of the original) vs. only around 200 hours in this work.

Response to the comments

Dear Reviewer,

First of all, many thanks for your comments on our manuscript. In the past two weeks, we have revised the manuscript carefully point-by-point based on your comments by supplementing new data about the fuel cell tests with higher Pt loading on both sides. Based on it, we have supplemented more analysis to the revised manuscript. More details can be found in the following response to your comments or in the revised manuscript highlighted in red.

Reviewers' comments:

Reviewer #1 (Remarks to the Author):

The authors addressed most of my questions properly. But I still think that durability of Pt₁-N is not so good, so I recommend to use 'good durability' in the manuscript instead of 'remarkable durability' (line 211).

Response to the comment: Dear reviewer, thanks for your suggestion here. Based on it, we have revised the "remarkable durability" to "good durability" in the manuscript (end of **Page-6**): "... indicating a good durability of Pt₁-N/BP as cathode in acidic fuel cells compared with other non-noble ORR catalysts (Supplementary Table 4)".

Reviewer #2 (Remarks to the Author):

The authors have addressed well all suggestions and comments. I think that it is publishable now.

Response to the comment: Dear Reviewer, many thanks for all your comments which helped lot for the improvement of this work.

Reviewer #3 (Remarks to the Author):

#1. I still think the claim of "super low Pt loading" and "high Pt utilization are misleading as I pointed out in my first Review that the ORR catalyst in this manuscript essentially belongs to a non-noble catalyst and as a result Pt utilization is not an appropriate characterization. For example, Fig 2 b's comparison is very misleading between Pt/C and a non-noble catalyst.

Response to the comment: Dear Reviewer, many thanks for your time on our manuscript. Based on your comment that our ORR catalyst " essentially belongs to a non-noble catalyst and as a result Pt utilization is not an appropriate characterization" in original Fig. 2b, then we removed the original Fig. 2b and Supplementary Fig. 10c about the comparison of Pt utilization between Pt/C and our catalyst in both acid and alkaline. To balance the Fig. 2, the tolerance of catalyst to CO and methanol in alkaline was moved from Fig. S10 to Fig.2 as Fig. 2e. Based on your comment, we also removed the claims of "super low Pt loading" and " high Pt utilization" relevant to the data (original Fig. 2b and Supplementary Fig. 10c about

the comparison of Pt utilization between Pt/C and our catalyst in both acid and alkaline) removed based on your comment here. (**Page5-7** in the revised manuscript).

#2. Another more misleading comparison is single cell performance in acid described in lines 191 and 207 referred to Fig 2f.. Everyone in this field knows that the state-of-the art commercial Pt/C is optimally employed for commercial CCMs (such as Ballard CCMs and Gore CCMs) with Pt loading ranging between 0.15 and 0.4 mgPt/cm² for the cathode side. Now the author manufactured a CCM with Pt/C commercial with only 0.04 mgPt/cm² and tested the cell performance with H₂/O₂ instead of H₂/air. I believe the author understands this loading and the H₂/O₂ are the best condition for his/her non-noble type catalyst and the non-noble type catalyst would be running into significant mass transfer issue due to the large thickness if the CCM were made with higher Pt loading and tested in H₂/air system.

Response to the comment: Thanks for all your comments. Based on it, in order to avoiding any possible misleading comparison, we have supplemented a test of fuel cell with higher Pt loadings on both sides (cathode: 0.2 mg_{Pt} cm⁻² and anode: 0.1 mg_{Pt} cm⁻²) to replace the previous one with low Pt loading. Indeed, a better fuel cell performance was obtained. For comparison, the new data were supplemented to the Fig. 2f in the revised manuscript. Based on all the new data and your comment, we have revised the discussion about the fuel cell results as shown in the following (or **Page6** in the revised manuscript):

" In order to further substantiate the high ORR performance of Pt₁-N/BP observed above, we performed acidic H₂/O₂ fuel cell tests with Pt₁-N/BP as cathode catalyst (SM). As shown in Fig. 2f (curves marked with ■), the Nafion-based acidic PEM H₂/O₂ fuel cell with Pt₁-N/BP as cathode (10 ugPt cm⁻²) and commercial Pt/C as anode (80 ugPt cm⁻²) possesses a high performance with maximum power density of 0.68 W cm⁻² at 80 °C, corresponding to a remarkable Pt utilization efficiency of 0.13g_{Pt}kW⁻¹ (Supplementary Table 3). For comparison, acidic PEM H₂/O₂ fuel cell (curves marked with ★) with commercial Pt/C as both cathode (200 ugPt cm⁻²) and anode (100 ugPt cm⁻²) was also tested at 80 °C. As expected, a much higher maximum power density of 1.02 W cm⁻² was observed; but it corresponds to a much lower Pt utilization efficiency of 0.29g_{Pt}kW⁻¹ due to a much higher Pt loading. Moreover, as shown in Fig. 2f, probably due to the mesoporous structure (Supplementary Fig. 11) and the much larger surface area (1102 m² g⁻¹) of Pt₁-N/BP than that of commercial Pt/C, the fuel cell with Pt₁-N/BP as cathode at low Pt loading of 10 ugPt cm⁻² shows performance as high as that with commercial Pt/C as cathode at a much higher Pt loading (200 ugPt cm⁻²) in the current density range < 0.6 A cm⁻², indicating a much higher Pt utilization of Pt single-atom-based Pt₁-N/BP than traditional Pt-NP-based Pt/C; while at higher current density, probably due to significant mass transfer issue in Pt₁-N/BP-based thick cathode, the performance of fuel cell with Pt₁-N/BP as cathode decays much faster than that with Pt/C as cathode."

#3. The durability result shown in the insert of Fig 2f is really not satisfying. People have shown the durability of commercial Pt/C in thousands of hours (before the voltage drops to 80% of the original) vs. only around 200 hours in this work.

Response to the comment: Thanks for all your comments. Indeed, as you mentioned, the long-term (thousands of hours) durability test was done before, but that was only for traditional commercial Pt/C with high Pt loading rather than non-noble catalysts.

While you also commented in your **#1** comment shown above that our ORR catalyst "essentially belongs to a non-noble catalyst", then it would be logically reasonable for us to compare the durability of our catalyst with other non-noble catalysts reported ever.

For this goal, we supplemented the durability comparison between our catalyst and other best non-noble catalysts tested under the same condition in fuel cell as shown in the following Table R1. Actually, Very few references have reported the durability tests of non-noble ORR catalysts in acidic H₂/O₂ fuel cells. The two results presented in Table R1 are the most relevant and best results that we found from the published references. From the table, one can see that the durability of our catalyst is not so bad even compared with the best non-noble catalysts reported by others.

Actually, based on the suggestion from the first reviewer, we have revised the "remarkable durability" to "good durability" in the manuscript (**in the last paragraph of Page6**: "...indicating a good durability of Pt₁-N/BP as cathode in acidic fuel cells compared with other non-noble ORR catalysts (Supplementary Table 4)"). To make it clearer, the following Table R1 was also supplemented to SI as **Supplementary Table 4**.

Table R1. Durability tests of catalysts in acidic H₂/O₂ fuel cell at fixed potential of 0.5 V

Samples	Current remained after a period of time at 0.5 V	References
Pt ₁ -N/BP	74% remained after 200 hours at 80°C	In this work
Pt ₁ -N/BP	90% remained after 200 hours at 70°C	In this work
S-Fe/N/C	28% remained after 100 hours at 80°C	Angew. Chem. Int. Ed. 2015, 54, 9907
Fe/N/C	44% remained after 100 hours at 80°C	Science 2009, 324, 71

REVIEWERS' COMMENTS:

Reviewer #2 (Remarks to the Author):

After carefully reading the reply, I found that the authors provided further testing data; in particular, changed the Pt loading of commercial catalysts. The results are convinced. Also, more comparisons have been carried out and put them in the supporting information. I am satisfied with them. I think that it is publishable now in NC.

Response to the comments from reviewers in red txt:

REVIEWERS' COMMENTS:

Reviewer #2 (Remarks to the Author):

After carefully reading the reply, I found that the authors provided further testing data; in particular, changed the Pt loading of commercial catalysts. The results are convinced. Also, more comparisons have been carried out and put them in the supporting information. I am satisfied with them. I think that it is publishable now in NC.

Response to the comment: Dear reviewer, many thanks for your time on our manuscript. You have helped lot for the improvement of this work.